# Differentiable Expected Hypervolume Improvement for Parallel Multi-Objective Bayesian Optimization

**Samuel Daulton**
Facebook
sdaulton@fb.com

**Maximilian Balandat**
Facebook
balandat@fb.com

**Eytan Bakshy**
Facebook
ebakshy@fb.com

## Abstract

In many real-world scenarios, decision makers seek to efficiently optimize multiple competing objectives in a sample-efficient fashion. Multi-objective Bayesian optimization (BO) is a common approach, but many of the best-performing acquisition functions do not have known analytic gradients and suffer from high computational overhead. We leverage recent advances in programming models and hardware acceleration for multi-objective BO using Expected Hypervolume Improvement (EHVI)—an algorithm notorious for its high computational complexity. We derive a novel formulation of $q$-Expected Hypervolume Improvement ($q$EHVI), an acquisition function that extends EHVI to the parallel, constrained evaluation setting. $q$EHVI is an exact computation of the joint EHVI of $q$ new candidate points (up to Monte-Carlo (MC) integration error). Whereas previous EHVI formulations rely on gradient-free acquisition optimization or approximated gradients, we compute exact gradients of the MC estimator via auto-differentiation, thereby enabling efficient and effective optimization using first-order and quasi-second-order methods. Our empirical evaluation demonstrates that $q$EHVI is computationally tractable in many practical scenarios and outperforms state-of-the-art multi-objective BO algorithms at a fraction of their wall time.

## 1 Introduction

The problem of optimizing multiple competing objectives is ubiquitous in scientific and engineering applications. For example in automobile design, an automaker will want to maximize vehicle durability and occupant safety, while using lighter materials that afford increased fuel efficiency and lower manufacturing cost [44, 72]. Evaluating the crash safety of an automobile design experimentally is expensive due to both the manufacturing time and the destruction of a vehicle. In such a scenario, sample efficiency is paramount. For a different example, video streaming web services commonly use adaptive control policies to determine the bitrate as the stream progresses in real time [47]. A decision maker may wish to optimize the control policy to maximize the quality of the video stream, while minimizing the stall time. Policy evaluation typically requires using the suggested policy on segments of live traffic, which is subject to opportunity costs. If long evaluation times are the limiting factor, multiple designs may be evaluated in parallel to significantly decrease end-to-end optimization time. For example, an automaker could manufacture multiple vehicle designs in parallel or a web service could deploy several control policies to different segments of traffic at the same time.

### 1.1 Background

**Multi-Objective Optimization:** In this work, we address the problem of optimizing a *vector-valued objective* $\boldsymbol{f}(\boldsymbol{x}) : R^d \to \mathbb{R}^M$ with $\boldsymbol{f}(\boldsymbol{x}) = \big(f^{(1)}(\boldsymbol{x}), ..., f^{(M)}(\boldsymbol{x})\big)$ over a bounded set $\mathcal{X} \subset \mathbb{R}^d$. We consider the scenario in which the $f^{(i)}$ are expensive-to-evaluate black-box functions with

no known analytical expression, and no observed gradients. *Multi-objective* (MO) optimization problems typically do not have a single best solution; rather, the goal is to identify the set of *Pareto optimal* solutions such that any improvement in one objective means deteriorating another. Without loss of generality, we assume the goal is to maximize all objectives. We say a solution $\boldsymbol{f}(\boldsymbol{x})$ *Pareto dominates* another solution $\boldsymbol{f}(\boldsymbol{x}')$ if $f^{(m)}(\boldsymbol{x}) \geq f^{(m)}(\boldsymbol{x}') \; \forall \; m = 1, \ldots, M$ and there exists $m' \in \{1, \ldots, M\}$ such that $f^{(m')}(\boldsymbol{x}) > f^{(m')}(\boldsymbol{x}')$. We write $\boldsymbol{f}(\boldsymbol{x}) \succ \boldsymbol{f}(\boldsymbol{x}')$. Let $\mathcal{P}^* = \{\boldsymbol{f}(\boldsymbol{x}) \; s.t. \; \nexists \; \boldsymbol{x}' \in \mathcal{X} \; : \; \boldsymbol{f}(\boldsymbol{x}') \succ \boldsymbol{f}(\boldsymbol{x})\}$ and $\mathcal{X}^* = \{\boldsymbol{x} \in \mathcal{X} \; s.t. \; \boldsymbol{f}(\boldsymbol{x}) \in \mathcal{P}^*\}$ denote the set of Pareto optimal solutions and Pareto optimal inputs, respectively. Provided with the Pareto set, decision-makers can select a solution with an objective trade-off according to their preferences.

A common approach for solving MO problems is to use evolutionary algorithms (e.g. NSGA-II), which are robust multi-objective optimizers, but require a large number of function evaluations [14]. Bayesian optimization (BO) offers a far more sample-efficient alternative [57].

**Bayesian Optimization:** BO [38] is an established method for optimizing expensive-to-evaluate black-box functions. BO relies on a probabilistic *surrogate model*, typically a Gaussian Process (GP) [55], to provide a posterior distribution $\mathbb{P}(\boldsymbol{f}|\mathcal{D})$ over the true function values $\boldsymbol{f}$ given the observed data $\mathcal{D} = \{(\boldsymbol{x}_i, \boldsymbol{y}_i)\}_{i=1}^n$. An *acquisition function* $\alpha : \mathcal{X}_{\text{cand}} \mapsto \mathbb{R}$ employs the surrogate model to assign a utility value to a set of candidates $\mathcal{X}_{\text{cand}} = \{\boldsymbol{x}_i\}_{i=1}^q$ to be evaluated on the true function. While the true $\boldsymbol{f}$ may be expensive-to-evaluate, the surrogate-based acquisition function is not, and can thus be efficiently optimized to yield a set of candidates $\mathcal{X}_{\text{cand}}$ to be evaluated on $\boldsymbol{f}$. If gradients of $\alpha(\mathcal{X}_{\text{cand}})$ are available, gradient-based methods can be utilized. If not, gradients are either approximated (e.g. with finite differences) or gradient-free methods (e.g. DIRECT [37] or CMA-ES [32]) are used.

## 1.2 Limitations of current approaches

In the single-objective (SO) setting, a large body of work focuses on practical extensions to BO for supporting *parallel* evaluation and outcome constraints [49, 30, 66, 25, 43]. Less attention has been given to such extensions in the MO setting. Moreover, the existing constrained and parallel MO BO options have limitations: 1) many rely on scalarizations to transform the MO problem into a SO one [40]; 2) many acquisition functions are computationally expensive to compute [52, 21, 6, 71]; 3) few have known analytical gradients or are differentiable [19, 62, 33]; 4) many rely on heuristics to extend sequential algorithms to the parallel setting [27, 62].

A natural acquisition function for MO BO is Expected Hypervolume Improvement (EHVI). Maximizing the hypervolume (HV) has been shown to produce Pareto fronts with excellent coverage [73, 12, 69]. However, there has been little work on EHVI in the parallel setting, and the work that has been done resorts to approximate methods [71, 28, 62]. A vast body of literature has focused on efficient EHVI computation [34, 20, 67], but the time complexity for computing EHVI is exponential in the number of objectives—in part due the hypervolume indicator itself incurring a time complexity that scales super-polynomially with the number of objectives [68]. Our core insight is that by exploiting advances in auto-differentiation and highly parallelized hardware [51], we can make EHVI computations fast and practical.

## 1.3 Contributions

In this work, we derive a novel formulation of the parallel $q$-Expected Hypervolume Improvement acquisition function ($q$EHVI) that is exact up to Monte-Carlo (MC) integration error. We compute the *exact* gradient of the MC estimator of $q$EHVI using auto-differentiation, which allows us to employ efficient and effective gradient-based optimization methods. Rather than using first-order gradient methods, we instead leverage the sample average approximation (SAA) approach from [5] to use higher-order deterministic optimization methods, and we prove theoretical convergence guarantees under the SAA approach. Our formulation of $q$EHVI is embarrassingly parallel, and despite its computational cost would achieve constant time complexity given infinite processing cores. We demonstrate that, using modern GPU hardware and computing exact gradients, optimizing $q$EHVI is faster than existing state-of-the art methods in many practical scenarios. Moreover, we extend $q$EHVI to support auxiliary outcome constraints, making it practical in many real-world scenarios. Lastly, we demonstrate how modern auto-differentiation can be used to compute exact gradients of analytic EHVI, which has never been done before for $M > 2$ objectives. Our empirical evaluation

shows that $q$EHVI outperforms state-of-the-art multi-objective BO algorithms while using only a fraction of their wall time.

## 2 Related Work

Yang et al. [69] is the only previous work to consider exact gradients of EHVI, but the authors only derive an analytical gradient for the unconstrained two-objective, sequential optimization setting. All other works either do not optimize EHVI (e.g. they use it for pre-screening candidates [18]), optimize it with gradient-free methods [68], or using approximate gradients [62]. In contrast, we use exact gradients and demonstrate that optimizing EHVI using this gradient information is far more efficient.

There are many alternatives to EHVI for MO BO. For example, ParEGO [40] and TS-TCH [50] randomly scalarize the objectives and use Expected Improvement [38] and Thompson Sampling [61], respectively. SMS-EGO [53] uses HV in a UCB-based acquisition function and is more scalable than EHVI [54]. ParEGO and SMS-EGO have only been considered for the $q = 1$, unconstrained setting. Predictive entropy search for MO BO (PESMO) [33] has been shown to be another competitive alternative and has been extended to handle constraints [26] and parallel evaluations [27]. MO max-value entropy search (MO-MES) has been shown to achieve superior optimization performance and faster wall times than PESMO, but is limited to $q = 1$.

Wilson et al. [65] empirically and theoretically show that *sequential greedy* selection of $q$ candidates achieves performance comparable to jointly optimizing $q$ candidates for many acquisition functions (including [63, 66]). The sequential greedy approach integrates over the posterior of the unobserved outcomes corresponding to the previously selected candidates in the $q$-batch. Sequential greedy optimization often yields better empirical results because the optimization problem has a lower dimension: $d$ in each step, rather than $qd$ in the joint problem. Most prior works in the MO setting use a sequential greedy approximation or heuristics [62, 71, 28, 10], but impute the unobserved outcomes with the posterior mean rather than integrating over the posterior [30]. For many joint acquisition functions involving expectations, this shortcut sacrifices the theoretical error bound on the sequential greedy approximation because the exact joint acquisition function over $\boldsymbol{x}_1, ..., \boldsymbol{x}_i$, $1 \leq i \leq q$ requires integration over the joint posterior $\mathbb{P}(\boldsymbol{f}(\boldsymbol{x}_1), ..., \boldsymbol{f}(\boldsymbol{x}_q)|\mathcal{D})$ and is not computed for $i > 1$.

Garrido-Merchán and Hernández-Lobato [27] and Wada and Hino [62] jointly optimize the $q$ candidates and, noting the difficulty of the optimization, both papers focus on deriving gradients to aid in the optimization. Wada and Hino [62] defined the $q$EHVI acquisition function, but after finding it challenging to optimize $q$ candidates jointly (without exact gradients), the authors propose optimizing an alternative acquisition function instead of exact $q$EHVI. In contrast, our novel $q$EHVI formulation allows for gradient-based parallel and sequential greedy optimization, with proper integration over the posterior for the latter.

Feliot et al. [22] and Abdolshah et al. [1] proposed extensions of EHVI to the constrained $q = 1$ setting, but neither considers the batch setting and both rely on gradient-free optimization.

## 3 Differentiable $q$-Expected Hypervolume Improvement

In this section, we review HVI and EHVI computation by means of box decompositions, and explain our novel formulation for the parallel setting.

**Definition 1.** *Given a reference point $\boldsymbol{r} \in \mathbb{R}^M$, the hypervolume indicator (HV) of a finite approximate Pareto set $\mathcal{P}$ is the $M$-dimensional Lebesgue measure $\lambda_M$ of the space dominated by $\mathcal{P}$ and bounded from below by $\boldsymbol{r}$: $\mathrm{HV}(\mathcal{P}, \boldsymbol{r}) = \lambda_M\big(\bigcup_{i=1}^{|\mathcal{P}|}[\boldsymbol{r}, \boldsymbol{y}_i]\big)$, where $[\boldsymbol{r}, \boldsymbol{y}_i]$ denotes the hyper-rectangle bounded by vertices $\boldsymbol{r}$ and $\boldsymbol{y}_i$.*

**Definition 2.** *Given a Pareto set $\mathcal{P}$ and reference point $\boldsymbol{r}$, the hypervolume improvement (HVI) of a set of points $\mathcal{Y}$ is: $\mathrm{HVI}(\mathcal{Y}, \mathcal{P}, \boldsymbol{r}) = \mathrm{HV}(\mathcal{P} \cup \mathcal{Y}, \boldsymbol{r}) - \mathrm{HV}(\mathcal{P}, \boldsymbol{r})$.*[1]

EHVI is the expectation of HVI over the posterior $\mathbb{P}(\boldsymbol{f}, \mathcal{D})$: $\alpha_{\mathrm{EHVI}}(\mathcal{X}_{\mathrm{cand}}) = \mathbb{E}\big[\mathrm{HVI}(\boldsymbol{f}(\mathcal{X}_{\mathrm{cand}}))\big]$. In the sequential setting, and assuming the objectives are independent and modeled with independent

GPs, EHVI can be expressed in closed form [69]. In other settings, EHVI can be approximated with MC integration. Following previous work, we assume that the reference point is known and specified by the decision maker [69] (see Appendix E.1.1 for additional discussion).

## 3.1 A review of hypervolume improvement computation using box decompositions

**Definition 3.** *For a set of objective vectors* $\{\boldsymbol{f}(\boldsymbol{x}_i)\}_{i=1}^q$, *a reference point* $\boldsymbol{r} \in \mathbb{R}^M$, *and a non-dominated set* $\mathcal{P}$, *let* $\Delta(\{\boldsymbol{f}(\boldsymbol{x}_i)\}_{i=1}^q, \mathcal{P}, \boldsymbol{r}) \subset \mathbb{R}^M$ *denote the set of points (i) are dominated by* $\{\boldsymbol{f}(\boldsymbol{x}_i)\}_{i=1}^q$, *dominate* $\boldsymbol{r}$, *and are not dominated by* $\mathcal{P}$.

Given $\mathcal{P}, \boldsymbol{r}$, the HVI of a new point $\boldsymbol{f}(\boldsymbol{x})$ is the HV of the intersection of space dominated by $\mathcal{P} \cup \{\boldsymbol{f}(\boldsymbol{x})\}$ and the non-dominated space. Figure 1b illustrates this for one new point $\boldsymbol{f}(\boldsymbol{x})$ for $M = 2$. The yellow region is $\Delta(\{\boldsymbol{f}(\boldsymbol{x})\}, \mathcal{P}, \boldsymbol{r})$ and the hypervolume improvement is the volume covered by $\Delta(\{\boldsymbol{f}(\boldsymbol{x})\}, \mathcal{P}, \boldsymbol{r})$. Since $\Delta(\{\boldsymbol{f}(\boldsymbol{x})\}, \mathcal{P}, \boldsymbol{r})$ is often a non-rectangular polytope, HVI is typically computed by partitioning the non-dominated space into disjoint axis-parallel rectangles [12, 68] (see Figure 1a) and using piece-wise integration [18].

Let $\{S_k\}_{k=1}^K$ be a partitioning the of non-dominated space into disjoint hyper-rectangles, where each $S_k$ is defined by a pair of lower and upper vertices $\boldsymbol{l}_k \in \mathbb{R}^M$ and $\boldsymbol{u}_k \in \mathbb{R}^M \cup \{\boldsymbol{\infty}\}$. The high level idea is to sum the HV of $S_k \cap \Delta(\{\boldsymbol{f}(\boldsymbol{x})\}, \mathcal{P}, \boldsymbol{r})$ over all $S_k$. For each hyper-rectangle $S_k$, the intersection of $S_k$ and $\Delta(\{\boldsymbol{f}(\boldsymbol{x})\}, \mathcal{P}, \boldsymbol{r})$ is a hyper-rectangle where the lower bound vertex is $\boldsymbol{l}_k$ and the upper bound vertex is the component-wise minimum of $\boldsymbol{u}_k$ and the new point $\boldsymbol{f}(\boldsymbol{x})$: $\boldsymbol{z}_k := \min[\boldsymbol{u}_k, \boldsymbol{f}(\boldsymbol{x})]$.

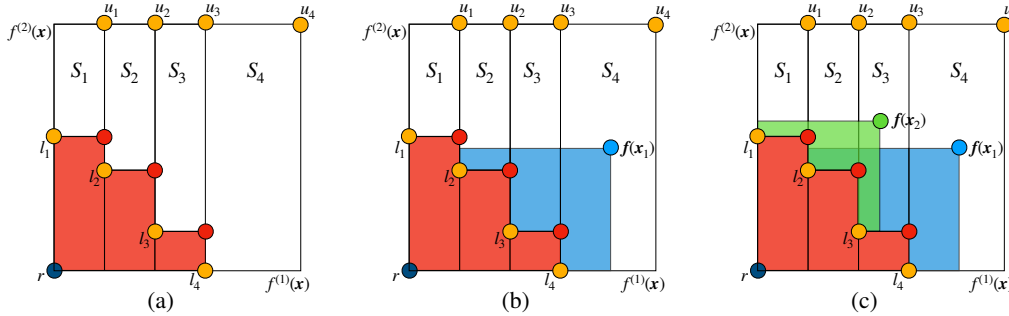

Figure 1: For M=2, (a) the dominated space (red) and the non-dominated space partitioned into disjoint boxes (white), (b) the HVI of one new point $\boldsymbol{f}(\boldsymbol{x})$, and (c) the HVI of two new points $\boldsymbol{f}(\boldsymbol{x}_1), \boldsymbol{f}(\boldsymbol{x}_2)$.

Hence, the HVI of a single outcome vector $\boldsymbol{f}(\boldsymbol{x})$ within $S_k$ is given by $\text{HVI}_k(\boldsymbol{f}(\boldsymbol{x}), \boldsymbol{l}_k, \boldsymbol{u}_k) = \lambda_M(S_k \cap \Delta(\{\boldsymbol{f}(\boldsymbol{x})\}, \mathcal{P}, \boldsymbol{r})) = \prod_{m=1}^M [z_k^{(m)} - l_k^{(m)}]_+$, where $u_k^{(m)}, l_k^{(m)}, f^{(m)}(\boldsymbol{x})$, and $z_k^{(m)}$ denote the $m^{\text{th}}$ component of the corresponding vector and $[\cdot]_+$ denotes the $\min(\cdot, 0)$ operation. Summing over rectangles yields

$$\text{HVI}(\boldsymbol{f}(\boldsymbol{x})) = \sum_{k=1}^K \text{HVI}_k(\boldsymbol{f}(\boldsymbol{x}), \boldsymbol{l}_k, \boldsymbol{u}_k) = \sum_{k=1}^K \prod_{m=1}^M [z_k^{(m)} - l_k^{(m)}]_+ \qquad (1)$$

## 3.2 Computing $q$-Hypervolume Improvement via the Inclusion-Exclusion Principle

Figure 1c illustrates the HVI in the $q = 2$ setting. Given $q$ new points $\{\boldsymbol{f}(\boldsymbol{x}_i)\}_{i=1}^q$, let $A_i := \Delta(\{\boldsymbol{f}(\boldsymbol{x}_i)\}, \mathcal{P}, \boldsymbol{r})$ for $i = 1, \ldots, q$ be the space dominated by $\boldsymbol{f}(\boldsymbol{x}_i)$ but not dominated by $\mathcal{P}$, independently of the other $q - 1$ points. Note that $\lambda_M(A_i) = \text{HVI}(\boldsymbol{f}(\boldsymbol{x}_i))$. The union of the subsets $A_i$ is the space dominated jointly by the $q$ new points: $\bigcup_{i=1}^q A_i = \bigcup_{i=1}^q \Delta(\{\boldsymbol{f}(\boldsymbol{x}_i)\}, \mathcal{P}, \boldsymbol{r})$, and the Lebesgue measure $\lambda_M(\bigcup_{i=1}^q A_i)$ is the joint HVI from the $q$ new points. Since each subspace $A_i$ is bounded, the restricted Lebesgue measure is finite and we may compute $\lambda_M(\bigcup_{i=1}^q A_i)$ using the inclusion-exclusion principle [13, 59]:

$$\text{HVI}(\{\boldsymbol{f}(\boldsymbol{x}_i)\}_{i=1}^q) = \lambda_M\left(\bigcup_{i=1}^q A_i\right) = \sum_{j=1}^q (-1)^{j+1} \sum_{1 \le i_1 \le \ldots \le i_j \le q} \lambda_M(A_{i_1} \cap \cdots \cap A_{i_j}) \qquad (2)$$

Since $\{S_k\}_{k=1}^K$ is a disjoint partition, $\lambda_M(A_{i_1} \cap \cdots \cap A_{i_j}) = \sum_{k=1}^K \lambda_M(S_k \cap A_{i_1} \cap \cdots \cap A_{i_j})$, we can compute $\lambda_M(A_{i_1} \cap \cdots \cap A_{i_j})$ in a piece-wise fashion across the $K$ hyper-rectangles $\{S_k\}_{k=1}^K$ as the HV of the intersection of $A_{i_1} \cap \cdots \cap A_{i_j}$ with each hyper-rectangle $S_k$. The inclusion-exclusion principle has been proposed for computing HV (not HVI) [45], but it is rarely used because complexity scales exponentially with the number of elements. However, the inclusion-exclusion principle is practical for computing the joint HVI of $q$ points since typically $q << |\mathcal{P}|$.

This formulation has three advantages. First, while the new dominated space $A_i$ can be a non-rectangular polytope, the intersection $A_i \cap S_k$ is a *rectangular* polytope, which simplifies computation of overlapping hypervolume. Second, the vertices defining the hyper-rectangle $S_k \cap A_{i_1} \cap \cdots \cap A_{i_j}$ are easily derived. The lower bound is simply the $l_k$ lower bound of $S_k$, and the upper bound is the component-wise minimum $z_{k,i_1,\ldots i_j} := \min\left[u_k, f(x_{i_1}), \ldots, f(x_{i_j})\right]$. Third, computation can be across all intersections of subsets $A_{i_1} \cap \cdots \cap A_{i_j}$ for $1 \le i_j \le \ldots \le i_j \le q$ and across all $K$ hyper-rectangles can be performed in parallel. Explicitly, the HVI is computed as:

$$\text{HVI}(\{f(x_i)\}_{i=1}^q) = \sum_{k=1}^K \sum_{j=1}^q \sum_{X_j \in \mathcal{X}_j} (-1)^{j+1} \prod_{m=1}^M \left[z_{k,X_j}^{(m)} - l_k^{(m)}\right]_+ \quad (3)$$

where $\mathcal{X}_j := \{X_j \subset \mathcal{X}_{\text{cand}} : |X_j| = j\}$ is the superset of all subsets of $\mathcal{X}_{\text{cand}}$ of size $j$, and $z_{k,X_j}^{(m)} := z_{k,i_1,\ldots i_j}^{(m)}$ for $X_j = \{x_{i_1}, \ldots, x_{i_j}\}$. See Appendix A for further details of the derivation.

### 3.3 Computing *Expected* $q$-Hypervolume Improvement

The above approach for computing HVI assumes that we know the true objective values $f(\mathcal{X}_{\text{cand}}) = \{f(x_i)\}_{i=1}^q$. In BO, we instead compute qEHVI as the expectation over the posterior model posterior:

$$\alpha_{q\text{EHVI}}(\mathcal{X}_{\text{cand}}) = \mathbb{E}\left[\text{HVI}(f(\mathcal{X}_{\text{cand}}))\right] = \int_{-\infty}^{\infty} \text{HVI}(f(\mathcal{X}_{\text{cand}}))df. \quad (4)$$

Since no known analytical form is known [70] for $q > 1$ (or in the case of correlated outcomes), we estimate (4) using MC integration with samples from the joint posterior $\{f_t(x_i)\}_{i=1}^q \sim \mathbb{P}(f(x_1), \ldots, f(x_q)|\mathcal{D}), t = 1, \ldots N$. Let $z_{k,X_j,t}^{(m)} := \min\left[u_k, \min_{x' \in X_j} f_t(x')\right]$. Then,

$$\hat{\alpha}_{q\text{EHVI}}^N(\mathcal{X}_{\text{cand}}) = \frac{1}{N}\sum_{t=1}^N \text{HVI}(f_t(\mathcal{X}_{\text{cand}})) = \frac{1}{N}\sum_{t=1}^N \sum_{k=1}^K \sum_{j=1}^q \sum_{X_j \in \mathcal{X}_j}(-1)^{j+1}\prod_{m=1}^M [z_{k,X_j,t}^{(m)} - l_k^{(m)}]_+ \quad (5)$$

Provided that $\{S_k\}_{k=1}^K$ is an exact partitioning, (5) is an *exact* computation of qEHVI up to the MC estimation error, which scales as $1/\sqrt{N}$ when using $iid$ MC samples regardless of the dimension of the search space [18]. In practice, we use randomized quasi MC methods [8] to reduce the variance and empirically observe low estimation error (see Figure 5a in the Appendix for a comparison of analytic EHVI and (quasi-)MC-based qEHVI).

qEHVI requires computing the volume of $2^q - 1$ hyper-rectangles (the number of subsets of q) for each of $K$ hyper-rectangles and $N$ MC samples. Given posterior samples, the time complexity on a single-threaded machine is: $T_1 = O(MNK(2^q - 1))$. In the two-objective case, $K = |\mathcal{P}| + 1$, but $K$ is super-polynomial in $M$ [68]. The number of boxes required for a decomposition of the non-dominated space is unknown for $M \ge 4$ [68]. qEHVI is agnostic to the partitioning algorithm used, and in F.4, we demonstrate using qEHVI in higher-dimensional objective spaces using an approximate box decomposition algorithm [11]. Despite the daunting workload, the critical work path—the time complexity of the smallest non-parallelizable unit—is constant: $T_\infty = O(1)$.[2] On highly-threaded many-core hardware (e.g. GPUs), our formulation achieves tractable wall times in many practical scenarios: as is shown in Figure 11 in the Appendix, the computation time is nearly constant with increasing $q$ until an inflection point at which the workload saturates the available cores. For additional discussion of both time and memory complexity of qEHVI see Appendix A.4.

### 3.4 Outcome Constraints

Our proposed qEHVI acquisition function is easily extended to constraints on auxiliary outcomes. We consider the scenario where we receive observations of $M$ objectives $f(x) \in \mathbb{R}^M$ and $V$ constraints

$\boldsymbol{c}^{(v)} \in \mathbb{R}^V$, all of which are assumed to be "black-box". We assume w.l.o.g. that $\boldsymbol{c}^{(v)}$ is feasible iff $\boldsymbol{c}^{(v)} \geq 0$. In the constrained optimization setting, we aim to identify the feasible Pareto set: $\mathcal{P}_{\text{feas}} = \{\boldsymbol{f}(\boldsymbol{x}) \ \ s.t. \ \ \boldsymbol{c}(\boldsymbol{x}) \geq \boldsymbol{0}, \ \nexists \ \boldsymbol{x}' : \boldsymbol{c}(\boldsymbol{x}') \geq \boldsymbol{0}, \ \boldsymbol{f}(\boldsymbol{x}') \succ \boldsymbol{f}(\boldsymbol{x})\}$. The natural improvement measure in the constrained setting is *feasible* HVI, which we define for a single candidate point $\boldsymbol{x}$ as $\text{HVI}_{\text{C}}(\boldsymbol{f}(\boldsymbol{x}), \boldsymbol{c}(\boldsymbol{x})) := \text{HVI}[\boldsymbol{f}(\boldsymbol{x})] \cdot \mathbb{1}[\boldsymbol{c}(\boldsymbol{x}) \geq \boldsymbol{0}]$. Taking expectations, the constrained expected HV can be seen to be the HV weighted by the probability of feasibility. In Appendix A.3, we detail how performing feasibility-weighting on the sample-level allows us to include such auxiliary outcome constraints into our MC formulation in a straightforward way.

# 4 Optimizing $q$-Expected Hypervolume Improvement

## 4.1 Differentiability

While an analytic formula for the gradient of EHVI exists for the $M = 2$ objective case in the unconstrained, sequential ($q = 1$) setting, no such formula is known in 1) the case of $M > 2$ objectives, 2) the constrained setting, and 3) for $q > 1$. Leveraging the re-parameterization trick [39, 64] and auto-differentiation, we are able to automatically compute exact gradients of the MC-estimator $q$EHVI in *all* of the above settings, as well as the gradient of analytic EHVI for $M \geq 2$ (see Figure 5b in the Appendix for a comparison of the exact gradients of EHVI and the sample average gradients of $q$EHVI for $M = 3$).[3,4]

## 4.2 Optimization via Sample Average Approximation

We show in Appendix C that if mean and covariance function of the GP are sufficiently regular, the gradient of the MC estimator (5) is an unbiased estimate of the gradient of the exact acquisition function (4). To maximize $q$EHVI, we could therefore directly apply stochastic optimization methods, as has previously been done for single-outcome acquisition functions [64, 66]. Instead, we opt to use the sample average approximation (SAA) approach from Balandat et al. [5], which allows us to employ deterministic, higher-order optimizers to achieve faster convergence rates. Informally (see Appendix C for the formal statement), if $\hat{x}_N^* \in \arg\max_{\boldsymbol{x} \in \mathcal{X}} \hat{\alpha}_{q\text{EHVI}}^N(\boldsymbol{x})$, we can show under some regularity conditions that, as $N \to \infty$, (i) $\hat{\alpha}_{q\text{EHVI}}^N(\hat{x}_N^*) \to \max_{x \in \mathcal{X}} \alpha_{q\text{EHVI}}(\boldsymbol{x}) \ \ a.s.$, and (ii) $\text{dist}(\hat{x}_N^*, \arg\max_{\boldsymbol{x} \in \mathcal{X}} \alpha_{q\text{EHVI}}(\boldsymbol{x})) \to 0 \ \ a.s.$. These results hold for any covariance function satisfying the regularity conditions, including such ones that model correlation between outcomes. In particular, our results do not require the outputs to be modeled by independent GPs.

Figure 2a demonstrates the importance of using exact gradients for efficiently and effectively optimizing EHVI and $q$EHVI by comparing the following optimization methods: L-BFGS-B with exact gradients, L-BFGS-B with gradients approximated via finite differences, and CMA-ES (without gradients). The cumulative time spent optimizing the acquisition function is an order of magnitude less when using exact gradients rather than approximate gradients or zeroth order methods.

## 4.3 Sequential Greedy and Joint Batch Optimization

Jointly optimizing $q$ candidates increases in difficulty with $q$ because the problem dimension is $dq$. An alternative is to sequentially and greedily select candidates and condition the acquisition function on the previously selected pending points when selecting the next point [65]. Using a submodularity argument similar to that in Wilson et al. [64], the sequential greedy approximation of $q$EHVI enjoys regret of no more than $\frac{1}{e}\alpha_{q\text{EHVI}}^*$, where $\alpha_{q\text{EHVI}}^*$ is the optima of $\alpha_{q\text{EHVI}}$ [23] (see Appendix B).

Although sequential greedy approaches have been considered for many acquisition functions [65], no previous work has proposed a proper sequential greedy approach (with integration over the posterior) for parallel EHVI, as this would require computing the Pareto front under each sample $\boldsymbol{f}_t$ from the joint posterior before computing the hypervolume improvement. These operations would be computationally expensive for even modest $N$ and non-differentiable. $q$EHVI avoids determining the Pareto set for each sample by using inclusion-exclusion principle to compute the joint HVI over the pending points $\boldsymbol{x}_1, ..., \boldsymbol{x}_{i-1}$ and the new candidate $\boldsymbol{x}_i$ for each MC sample. Figure 2b empirically

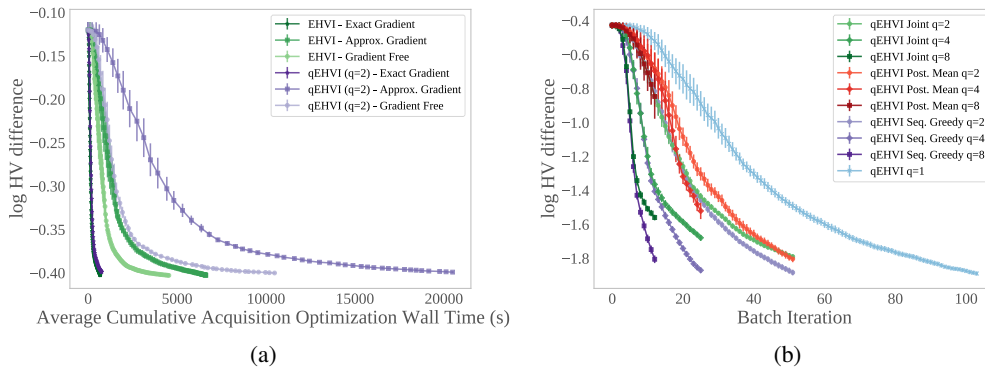

Figure 2: (a) A comparison of EHVI and $q$EHVI ($q = 2$) optimized with L-BFGS-B using exact gradients, L-BFGS-B using gradients approximated using finite differences, and CMA-ES, a gradient-free method. (b) A comparison of joint optimization, sequential greedy optimization with proper integration at the pending points, and sequential greedy using the posterior mean. Both plots show optimization performance on a DTLZ2 problem ($d = 6, M = 2$) with a budget of 100 evaluations (plus the initial quasi-random design). We report means and 2 standard errors across 20 trials.

demonstrates the improved optimization performance from properly integrating over the unobserved outcomes rather than using the posterior mean or jointly optimizing the $q$ candidates.

## 5 Benchmarks

We empirically evaluate $q$EHVI on synthetic and real world optimization problems. We compare $q$EHVI[5] against existing state-of-the-art methods including SMS-EGO[6], PESMO[6], TS-TCH[5], and analytic EHVI [68] with gradients[5]. Additionally, we compare against a novel extension of ParEGO [40] that supports parallel evaluation and constraints (neither of which have been done before to our knowledge); we call this method $q$PAREGO[5]. Additionally, we include a quasi-random baseline that selects candidates from a scrambled Sobol sequence. See Appendix E.1 for details on all baseline algorithms.

**Synthetic Benchmarks**   We evaluate optimization performance on four benchmark problems in terms of log hypervolume difference, which is defined as the difference between the hypervolume of the true (feasible) Pareto front and the hypervolume of the approximate (feasible) Pareto front based on the observed data; in the case that the true Pareto front is unknown (or not easily approximated), we evaluate the hypervolume indicator. All references points and search spaces are provided in Appendix E.2. For synthetic problems, we consider the Branin-Currin problem ($d = 2, M = 2$, convex Pareto front) [6] and the C2-DTLZ2 ($d = 12, M = 2, V = 1$, concave Pareto front), which is a standard constrained benchmark from the MO literature [16] (see Appendix F.1 for additional synthetic benchmarks).

**Real-World Benchmarks** *Structural Optimization in Automobile Safety Design* (VEHICLESAFETY): Vehicle crash safety is an important consideration in the structural design of automobiles. A lightweight car is preferable because of its potentially lower manufacturing cost and better fuel economy, but lighter material can fare worse than sturdier alternatives in a collision, potentially leading to increased vehicle damage and more severe injury to the vehicle occupants [72]. We consider the problem designing the thickness of 5 reinforced parts of the frontal frame of a vehicle that considerably affect crash safety. The goal is to minimize: 1) the *mass* of the vehicle; 2) the *collision acceleration* in a full frontal crash—a proxy for bio-mechanical trauma to the vehicle occupants from the acceleration; and 3) the *toe-board intrusion*—a measure of the most extreme

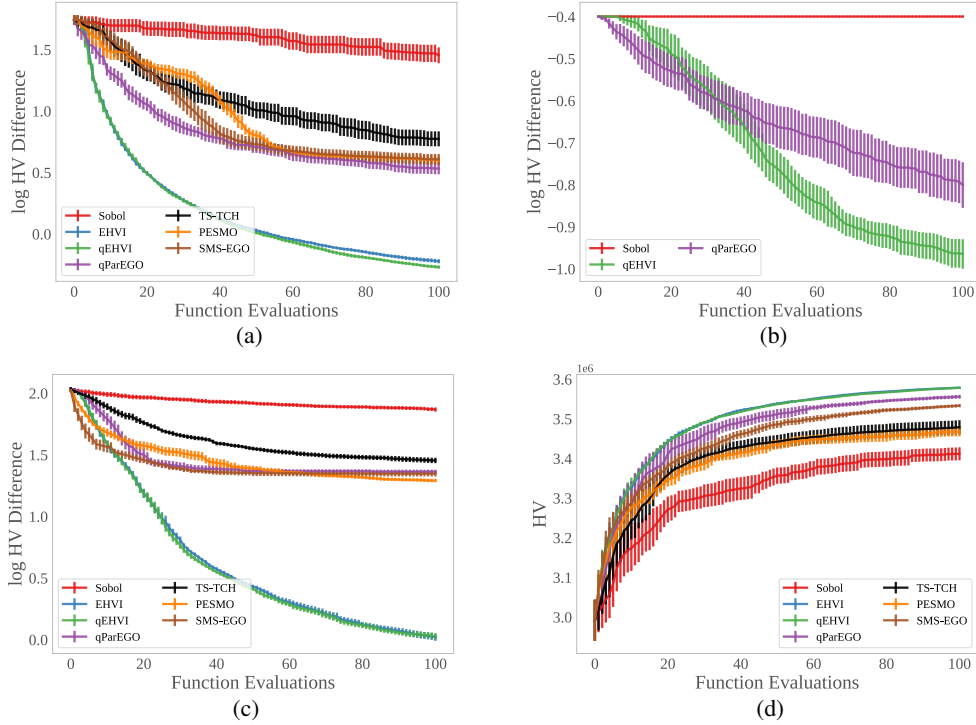

Figure 3: Sequential optimization performance on (a) on the Branin-Currin problem ($q = 1$), (b) the C2-DTLZ2 problem, (c) the vehicle crash safety problem ($q = 1$), and (d) the ABR control problem ($q = 1$). We report the means and 2 standard errors across 20 trials.

mechanical damage to the vehicle in an off-frontal collision [44]. For this problem, we optimize the surrogate from Tanabe and Ishibuchi [60].

*Policy Optimization for Adaptive Bitrate Control* (ABR): Many web services adapt video playback quality adaptively based on the receiver's network bandwith to maintain steady, high quality stream with minimal stalls and buffer periods [47]. Previous works have proposed controllers with different scalarized objective functions [46], but in many cases, engineers may prefer to learn the set of optimal trade-offs between their metrics of interest, rather than specifying a scalarized objective in advance. In this problem, we decompose the objective function proposed in Mao et al. [46] into its constituent metrics and optimize 4 parameters of an ABR control policy on the Park simulator [48] to maximize video quality (bitrate) and minimize stall time. See Appendix E.2 for details.

## 5.1 Results

Figure 3 shows that $q$EHVI outperforms all baselines in terms of sequential optimization performance on all evaluated problems. Table 1 shows that $q$EHVI achieves wall times that are an order of magnitude smaller than those of PESMO on a CPU in sequential optimization, and maintains competitive wall times even relative to $q$PAREGO (which has a significantly smaller workload) for large $q$ on a GPU. TS-TCH has by far the fastest wall time, but this comes at the cost of inferior optimization performance.

Figure 4 illustrates optimization performance of *parallel* acquisition functions for varying batch sizes. Increasing the level of parallelism leads to faster convergence for all algorithms (Figure 4a). In contrast with other algorithms, $q$EHVI's sample complexity does not deteriorate substantially when high levels of parallelism are used (Figure 4b).

Table 1: Acquisition Optimization wall time in seconds on a CPU (2x Intel Xeon E5-2680 v4 @ 2.40GHz) and a GPU (Tesla V100-SXM2-16GB). We report the mean and 2 standard errors across 20 trials. NA indicates that the algorithm does not support constraints.

| CPU | BRANINCURRIN | C2DTLZ2 | ABR | VEHICLESAFETY |
|---|---|---|---|---|
| PESMO ($q$=1) | 249.16 ($\pm$19.35) | NA | 214.16 ($\pm$18.38) | 492.64 ($\pm$58.98) |
| SMS-EGO ($q$=1) | 146.1 ($\pm$8.57) | NA | 89.54 ($\pm$5.79) | 115.11 ($\pm$8.21) |
| TS-TCH ($q$=1) | 2.82 ($\pm$0.03) | NA | 17.22 ($\pm$0.04) | 47.46 ($\pm$0.05) |
| qPAREGO ($q$=1) | 1.56 ($\pm$0.16) | 4.01 ($\pm$0.77) | 7.47 ($\pm$0.67) | 1.74 ($\pm$0.27) |
| EHVI ($q$=1) | 3.04 ($\pm$0.16) | NA | 2.48 ($\pm$0.19) | 15.18 ($\pm$2.24) |
| qEHVI ($q$=1) | 3.63 ($\pm$0.23) | 5.4 ($\pm$1.18) | 6.15 ($\pm$0.71) | 67.54 ($\pm$10.45) |

| GPU | BRANINCURRIN | C2DTLZ2 | ABR | VEHICLESAFETY |
|---|---|---|---|---|
| TS-TCH ($q$=1) | 0.07 ($\pm$0.00) | NA | 0.16 ($\pm$0.00) | 0.32 ($\pm$0.0) |
| TS-TCH ($q$=2) | 0.07 ($\pm$0.00) | NA | 0.15 ($\pm$0.00) | 0.34 ($\pm$0.01) |
| TS-TCH ($q$=4) | 0.09 ($\pm$0.01) | NA | 0.15 ($\pm$0.00) | 0.31 ($\pm$0.01) |
| TS-TCH ($q$=8) | 0.08 ($\pm$0.00) | NA | 0.16 ($\pm$0.00) | 0.34 ($\pm$0.01) |
| qPAREGO ($q$=1) | 3.2 ($\pm$0.37) | 3.85 ($\pm$0.91) | 9.64 ($\pm$0.96) | 3.44 ($\pm$0.51) |
| qPAREGO ($q$=2) | 7.12 ($\pm$0.81) | 12.1 ($\pm$2.77) | 21.19 ($\pm$1.53) | 7.32 ($\pm$0.97) |
| qPAREGO ($q$=4) | 15.34 ($\pm$1.69) | 39.71 ($\pm$7.40) | 35.46 ($\pm$2.32) | 17.2 ($\pm$2.29) |
| qPAREGO ($q$=8) | 32.11 ($\pm$4.14) | 99.58 ($\pm$15.20) | 72.52 ($\pm$5.04) | 39.72 ($\pm$7.13) |
| EHVI ($q$=1) | 4.53 ($\pm$0.23) | NA | 6.82 ($\pm$0.55) | 8.95 ($\pm$0.64) |
| qEHVI ($q$=1) | 5.98 ($\pm$0.28) | 3.36 ($\pm$0.94) | 7.71 ($\pm$0.67) | 10.43 ($\pm$0.64) |
| qEHVI ($q$=2) | 11.37 ($\pm$0.56) | 21.56 ($\pm$3.45) | 18.32 ($\pm$1.48) | 17.67 ($\pm$1.54) |
| qEHVI ($q$=4) | 25.29 ($\pm$1.51) | 89.18 ($\pm$10.86) | 44.44 ($\pm$3.53) | 54.25 ($\pm$4.17) |
| qEHVI ($q$=8) | 102.46 ($\pm$9.22) | 215.74 ($\pm$15.85) | 100.64 ($\pm$7.22) | 255.72 ($\pm$23.73) |

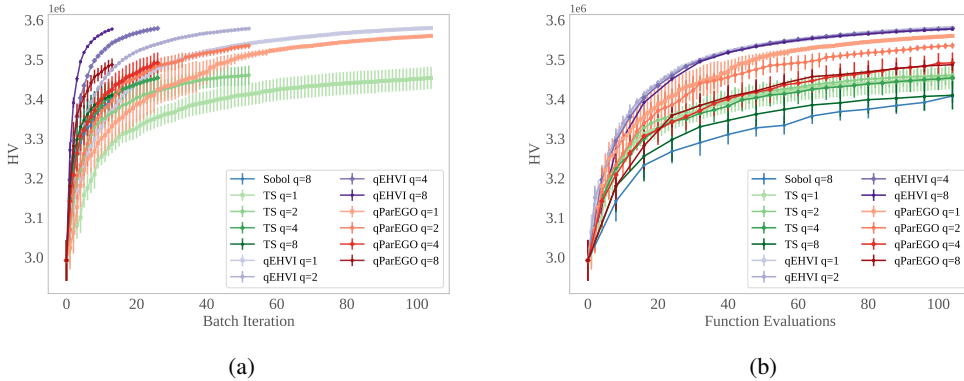

(a)  (b)

Figure 4: Parallel optimization performance on the ABR problem with varying batch sizes ($q$) by (a) *batch BO iterations* and (b) *function evaluations*.

## 6 Discussion

We present a practical and efficient acquisition function, $q$EHVI, for parallel, constrained multi-objective Bayesian optimization. Leveraging differentiable programming, modern parallel hardware, and the Sample Average Approximation, we efficiently optimize $q$EHVI via quasi second-order methods and provide theoretical convergence guarantees for our approach. Empirically, we demonstrate that our method out-performs state-of-the-art multi-objective Bayesian optimization methods.

One limitation of our approach is that it currently assumes noiseless observations, which, to our knowledge, is the case with all formulations of EHVI. Integrating over the uncertainty around the previous observations [43] by using MC samples over the new candidates and the training points, one may be able to account for the noise. Another limitation of $q$EHVI is that its scalability is limited the partitioning algorithm, precluding its use in high-dimensional objective spaces. More scalable partitioning algorithms, either approximate algorithms (e.g. the algorithm proposed by Couckuyt et al. [11], which we examine briefly in Appendix F.4) or more efficient exact algorithms that result in fewer disjoint hyper-rectangles (e.g. [41, 17, 69]), will improve the scalability and computation time of of $q$EHVI. We hope this work encourages researchers to consider more improvements from applying modern computational paradigms and tooling to Bayesian optimization.

# 7 Statement of Broader Impact

Optimizing a single outcome commonly comes at the expense of other secondary outcomes. In some cases, decision makers may be able to form a scalarization of their objectives in advance, but in the researcher's experience, formulating such trade-offs in advance is difficult for most. Improvements to the optimization performance and practicality of multi-objective Bayesian optimization have the potential to allow decision makers to better understand and make more informed decisions across multiple trade-offs. We expect these directions to be particularly important as Bayesian optimization is increasingly used for applications such as recommender systems [42], where auxiliary goals such as fairness must be accounted for. Of course, at the end of the day, exactly what objectives decision makers choose to optimize, and how they balance those trade-offs (and whether that is done in equitable fashion) is up to the individuals themselves.

## Acknowledgments

We would like to thank Daniel Jiang for helpful discussions around our theoretical results.

## Footnotes

[1]In this work, we omit the arguments $\mathcal{P}$ and $\boldsymbol{r}$ when referring to HVI for brevity.

[2]As evident from (5), the critical path consists of 3 multiplications and 5 summations.

[3]Technically, min and max are only sub-differentiable, but are known to be well-behaved [64]. In our MC setting with GP posteriors, $q$EHVI is differentiable w.p. 1 if $\boldsymbol{x}$ contains no repeated points.

[4]For the constrained case, we replace the indicator with a differentiable sigmoid approximation.

[5]Acquisition functions are available as part of the open-source library BoTorch [5]. Code is available at https://github.com/pytorch/botorch.

[6]We leverage existing implementations from the Spearmint library. The code is available at https://github.com/HIPS/Spearmint/tree/PESM.

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
