[Supplementary Material]

# Appendix to:

# Differentiable Expected Hypervolume Improvement for Parallel Multi-Objective Bayesian Optimization

## A  Derivation of $q$-Expected Hypervolume Improvement

### A.1  Hypervolume Improvement via the Inclusion-Exclusion Principle

The hypervolume improvement of $\boldsymbol{f}(\boldsymbol{x})$ within the hyper-rectangle $S_k$ is the volume of $S_k \cap \Delta(\{\boldsymbol{f}(\boldsymbol{x})\}, \mathcal{P}, \boldsymbol{r})$ and is given by:

$$\mathrm{HVI}_k\big(\boldsymbol{f}(\boldsymbol{x}), \boldsymbol{l}_k, \boldsymbol{u}_k\big) = \lambda_M\big(S_k \cap \Delta(\{\boldsymbol{f}(\boldsymbol{x})\}, \mathcal{P}, \boldsymbol{r})\big) = \prod_{m=1}^{M} \big[z_k^{(m)} - l_k^{(m)}\big]_+,$$

where $u_k^{(m)}, l_k^{(m)}, f^{(m)}(\boldsymbol{x})$, and $z_k^{(m)}$ denote the $m^{\text{th}}$ component of the corresponding vector and $[\cdot]_+$ denotes the $\min(\cdot, 0)$ operation. Summing over all $S_k$ gives the total hypervolume improvement:

$$\begin{aligned} \mathrm{HVI}\big(\boldsymbol{f}(\boldsymbol{x})\big) &= \sum_{k=1}^{K} \mathrm{HVI}_k\big(\boldsymbol{f}(\boldsymbol{x}), \boldsymbol{l}_k, \boldsymbol{u}_k\big) \\ &= \sum_{k=1}^{K} \lambda_M\big(S_k \cap \Delta(\{\boldsymbol{f}(\boldsymbol{x})\}, \mathcal{P}, \boldsymbol{r})\big) \\ &= \sum_{k=1}^{K} \prod_{m=1}^{M} \big[z_k^{(m)} - l_k^{(m)}\big]_+. \end{aligned}$$

We can extend the HVI computation to the $q > 1$ case using the inclusion-exclusion principle.

**Principle 1.** ***The inclusion-exclusion principle*** [13, 59, 9] *Given a finite measure space* $(B, \mathcal{A}, \mu)$ *and a finite sequence of potentially empty or overlapping sets* $\{A_i\}_i = 1^n$ *where* $A_i \in \mathcal{A}$ *and* $\mu(B) < \infty$, *then,*

$$\lambda_M\left(\bigcup_{i=1}^{p} A_i\right) = \sum_{j=1}^{p} (-1)^{j+1} \sum_{1 \le i_1 \le \ldots \le i_j \le p} \lambda_M\big(A_{i_1} \cap \ldots \cap A_{i_j}\big)$$

In the context of computing the joint HVI of $q$ new points $\{\boldsymbol{f}(\boldsymbol{x}_i)\}_{i=1}^{q}$, each subset $A_i$ for $i = 1, \ldots, q$ is the set of points contained in $\Delta(\{\boldsymbol{f}(\boldsymbol{x}_i)\}, \mathcal{P}, \boldsymbol{r})$ — independently of the other $q-1$ points. $\lambda_M(A_i)$ is the hypervolume improvement from the new point $\boldsymbol{f}(\boldsymbol{x}_i)$: $\lambda_M(A_i) = \mathrm{HVI}(\boldsymbol{f}(\boldsymbol{x}_i))$. The union of these subsets is the set of points in the new space dominated by the $q$ new points: $\bigcup_{i=1}^{q} A_i = \bigcup_{i=1}^{q} \Delta(\{\boldsymbol{f}(\boldsymbol{x}_i)\}, \mathcal{P}, \boldsymbol{r})$. The hypervolume of $\bigcup_{i=1}^{q} \Delta(\{\boldsymbol{f}(\boldsymbol{x}_i)\}, \mathcal{P}, \boldsymbol{r})$ is the hypervolume improvement from the $q$ new points:

$$\begin{aligned} \mathrm{HVI}(\{\boldsymbol{f}(\boldsymbol{x}_i)\}_{i=1}^{q}) &= \lambda_M\left(\bigcup_{i=1}^{q} A_i\right) \\ &= \sum_{j=1}^{q} (-1)^{j+1} \sum_{1 \le i_1 \le \ldots \le i_j \le q} \lambda_M\big(A_{i_1} \cap \cdots \cap A_{i_j}\big) \end{aligned}$$

To compute $\lambda_M(A_{i_1} \cap \cdots \cap A_{i_j})$, we partition the space covered by $A_{i_1} \cap \cdots \cap A_{i_j}$ across the $K$ hyper-rectangles $\{S_k\}_{k=1}^{K}$ and compute the hypervolume of the overlapping space of $A_{i_1} \cap \cdots \cap A_{i_j}$ with each $S_k$ independently. Since $\{S_k\}_{k=1}^{K}$ is a disjoint partition, summing over $K$ gives the hypervolume of $A_{i_1} \cap \cdots \cap A_{i_j}$:

$$\lambda_M\big(A_{i_1} \cap \cdots \cap A_{i_j}\big) = \sum_{k=1}^{K} \lambda_M\big(S_k \cap A_{i_1} \cap \cdots \cap A_{i_j}\big)$$

This has two advantages. First, the new dominated space $A_i$ can be a non-rectangular polytope, but the intersection $A_i \cap S_k$ is a *rectangular* polytope, which simplifies computation of overlapping hypervolume.

Second, the vertices defining the hyper-rectangle encapsulated by $S_k \cap A_{i_1} \cap \cdots \cap A_{i_j}$ are easily derived. The lower bound is simply the $\boldsymbol{l}_k$ lower bound of $S_k$ and the upper bound is the component-wise minimum $\boldsymbol{z}_{k,i_1,\dots i_j} = \min\left[\boldsymbol{u}_k, \boldsymbol{f}(\boldsymbol{x}_{i_1}), \dots, \boldsymbol{f}(\boldsymbol{x}_{i_j})\right]$.

Importantly, this is computationally tractable because this specific approach enables parallelizing computation across all intersections of subsets $A_{i_1} \cap \cdots \cap A_{i_j}$ for $1 \le i_j \le \dots \le i_j \le q$ and across all $K$ hyper-rectangles. Explicitly, the HVI is computed as:

$$
\begin{aligned}
\mathrm{HVI}(\{\boldsymbol{f}(\boldsymbol{x}_i)\}_{i=1}^q) &= \lambda_M\left(\bigcup_{i=1}^p A_i\right) \\
&= \sum_{j=1}^q \sum_{1 \le i_1 \le \dots \le i_j \le q} (-1)^{j+1} \lambda_M\left(A_{i_1} \cap \cdots \cap A_{i_j}\right) \\
&= \sum_{k=1}^K \sum_{j=1}^q \sum_{1 \le i_1 \le \dots \le i_j \le q} (-1)^{j+1} \lambda_M\left(S_k \cap A_{i_1} \cap \cdots \cap A_{i_j}\right) \\
&= \sum_{k=1}^K \sum_{j=1}^q \sum_{1 \le i_1 \le \dots \le i_j \le q} (-1)^{j+1} \lambda_M\left(S_k \cap \Delta(\{\boldsymbol{f}(\boldsymbol{x}_{i_1})\}, \mathcal{P}, \boldsymbol{r}) \cap \dots \cap \Delta(\{\boldsymbol{f}(\boldsymbol{x}_{i_j})\}, \mathcal{P}, \boldsymbol{r})\right) \\
&= \sum_{k=1}^K \sum_{j=1}^q \sum_{1 \le i_1 \le \dots \le i_j \le q} (-1)^{j+1} \prod_{m=1}^M \left[z_{k,i_1,\dots i_j}^{(m)} - l_k^{(m)}\right]_+ \\
&= \sum_{k=1}^K \sum_{j=1}^q \sum_{X_j \in \mathcal{X}_j} (-1)^{j+1} \prod_{m=1}^M \left[z_{k,X_j}^{(m)} - l_k^{(m)}\right]_+
\end{aligned}
$$

where $X_j$ is the superset all subsets of $\mathcal{X}_{\mathrm{cand}}$ of size $j$: $\mathcal{X}_j = \{X_j \subset \mathcal{X}_{\mathrm{cand}} : |X_j| = j\}$ and $z_{k,X_j}^{(m)} = z_{k,i_1,\dots i_j}^{(m)}$ for $X_j = \{\boldsymbol{x}_{i_1}, \dots, \boldsymbol{x}_{i_j}\}$.

### A.2  Computing *Expected* Hypervolume Improvement

The above approach for computing HVI assumes we know the true objective values $\{\boldsymbol{f}(\boldsymbol{x}_i)\}_{i=1}^q$. Since we do not know the true function values $\{\boldsymbol{f}(\boldsymbol{x}_i)\}_{i=1}^q$, we compute qEHVI as the expectation over the GP posterior.

$$
\alpha_{q\mathrm{EHVI}} = \mathbb{E}\left[\mathrm{HVI}(\{\boldsymbol{f}(\boldsymbol{x}_i)\}_{i=1}^q)\right] = \int_{\mathbb{R}^M} \mathrm{HVI}(\{\boldsymbol{f}(\boldsymbol{x}_i)\}_{i=1}^q) d\boldsymbol{f} \tag{6}
$$

In the sequential setting and under the assumption of independent outcomes, qEHVI is simply EHVI and can be expressed in closed form [69]. However when $q > 1$, there is no known analytical formulation [70]. Instead, we estimate the expectation in (6) using MC integration with samples from the joint posterior $\mathbb{P}(\boldsymbol{f}(\boldsymbol{x}_1), \dots, \boldsymbol{f}(\boldsymbol{x}_q)|\mathcal{D})$:

$$
\alpha_{q\mathrm{EHVI}} = \mathbb{E}\left[\mathrm{HVI}(\{\boldsymbol{f}(\boldsymbol{x}_i)\}_{i=1}^q)\right] \approx \frac{1}{N} \sum_{t=1}^N \mathrm{HVI}(\{\boldsymbol{f}_t(\boldsymbol{x}_i)\}_{i=1}^q) \tag{7}
$$

$$
= \frac{1}{N} \sum_{t=1}^N \sum_{k=1}^K \sum_{j=1}^q \sum_{X_j \in \mathcal{X}_j} (-1)^{j+1} \prod_{m=1}^M \left[\boldsymbol{z}_{k,X_j,t}^{(m)} - l_k^{(m)}\right]_+ \tag{8}
$$

where $\{\boldsymbol{f}_t(\boldsymbol{x}_i)\}_{i=1}^q \sim \mathbb{P}(\boldsymbol{f}(\boldsymbol{x}_1), \dots, \boldsymbol{f}(\boldsymbol{x}_q)|X, Y)$ is the $t^{\mathrm{th}}$ sample from the joint posterior over $\mathcal{X}_{\mathrm{cand}}$ and $\boldsymbol{z}_{k,X_j,t}^{(m)} = \min\left[\boldsymbol{u}_k, \min_{\boldsymbol{x}' \in X_j} \boldsymbol{f}_t(\boldsymbol{x}')\right]$.

### A.3  Supporting Outcome Constraints

Recall that we defined the constrained hypervolume improvement as

$$
\mathrm{HVI}_{\mathrm{c}}(\boldsymbol{f}(\boldsymbol{x}), \boldsymbol{c}(\boldsymbol{x})) = \mathrm{HVI}[\boldsymbol{f}(\boldsymbol{x})] \cdot \mathbb{1}[\boldsymbol{c}(\boldsymbol{x}) \ge \boldsymbol{0}]. \tag{9}
$$

For $q = 1$ and assuming independence of the objectives and the constraints, the expected $\mathrm{HVI}_{\mathrm{c}}$ is the product of the expected HVI and the probability of feasibility (the expectation of $\mathbb{1}[\boldsymbol{c}(\boldsymbol{x}) \ge \boldsymbol{0}]$) [22]. However, requiring objectives and constraints to be independent is unnecessary when estimating the expectation with MC integration using samples from the joint posterior.

In the parallel setting, if all constraints are satisfied for all $q$ candidates $\mathcal{X}_{\text{cand}} = \{\boldsymbol{x}_i\}_{i=1}^q$, $\text{HVI}_{\text{C}}$ is simply HVI. If a subset $\mathcal{V} \subset \mathcal{X}_{\text{cand}}$, $\mathcal{V} \neq \varnothing$ of the candidates violate at least one of the constraints, then the feasible HVI is the HVI of the set of feasible candidates: $\text{HVI}_{\text{C}}(\mathcal{X}_{\text{cand}}) = \text{HVI}(\mathcal{X}_{\text{cand}} \setminus \mathcal{V})$. That is, the *hypervolume contribution* (i.e. the marginal HVI) of an infeasible point is zero. In our formulation, HVI can be computed by multiplying (5) with an additional factor $\prod_{\boldsymbol{x}' \in X_j} \prod_{v=1}^V \mathbb{1}[c^{(v)}(\boldsymbol{x}') \geq 0]$:

$$\text{HVI}_{\text{C}}(\{\boldsymbol{f}(\boldsymbol{x}_i), \boldsymbol{c}(\boldsymbol{x}_i)\}_{i=1}^q) = \sum_{k=1}^K \sum_{j=1}^q \sum_{X_j \in \mathcal{X}_j} (-1)^{j+1} \left[ \left( \prod_{m=1}^M \left[ \boldsymbol{z}_{k,X_j}^{(m)} - l_k^{(m)} \right]_+ \right) \prod_{\boldsymbol{x}' \in X_j} \prod_{v=1}^V \mathbb{1}[c^{(v)}(\boldsymbol{x}') \geq 0] \right].$$
(10)

The additional factor $\prod_{\boldsymbol{x}' \in X_j} \prod_{v=1}^V \mathbb{1}[c^{(v)}(\boldsymbol{x}_a) \geq 0]$ indicates whether all constraints are satisfied for all candidates in a given subset $X_j$. Thus $\text{HVI}_{\text{C}}$ can be computed in the same fashion as HVI, but with the additional step of setting the HV of all subsets containing $\boldsymbol{x}'$ to zero if $\boldsymbol{x}'$ violates any constraint. We can now again perform MC integration as in (5) to compute the expected constrained hypervolume improvement.

In this formulation, the marginal hypervolume improvement from a candidate is weighted by the probability that the candidate is feasible. The marginal hypervolume improvements are highly dependent on the outcomes of the other candidates. Importantly, the MC-based approach enables us to properly estimate the marginal hypervolume improvements across candidates by sampling from the joint posterior.

Note that while the *expected* constrained hypervolume $\mathbb{E}\big[\text{HVI}_{\text{C}}(\{\boldsymbol{f}(\boldsymbol{x}_i), \boldsymbol{c}(\boldsymbol{x}_i)\}_{i=1}^q)\big]$ is differentiable, we may *not* differentiate inside the expectation (hence we cannot expect simply differentiating (10) on the sample-level to provide proper gradients). We therefore replace the indicator with a sigmoid function with temperature parameter $\epsilon$, which provides a differentiable relaxation

$$\mathbb{1}[c^{(v)}(\boldsymbol{x}') \geq 0] \approx s(c^{(v)}(\boldsymbol{x}'); \epsilon) := \frac{1}{1 + \exp(-c^{(v)}(\boldsymbol{x}')/\epsilon)}$$
(11)

that becomes exact in the limit $\epsilon \searrow 0$.

As in the unconstrained parallel scenario, there is no known analytical expression for the expected feasible hypervolume improvement. Therefore, we again use MC integration to approximate the expectation:

$$\alpha_{q\text{EHVI}_{\text{C}}}(\boldsymbol{x}) = \mathbb{E}\Big[\text{HVI}_{\text{C}}(\{\boldsymbol{f}(\boldsymbol{x}_i), \boldsymbol{c}(\boldsymbol{x}_i)\}_{i=1}^q)\Big] \tag{12a}$$

$$\approx \frac{1}{N} \sum_{t=1}^N \text{HVI}_{\text{C}}(\{\boldsymbol{f}_t(\boldsymbol{x}_i), c_t(\boldsymbol{x}_i)\}_{i=1}^q) \tag{12b}$$

$$\approx \frac{1}{N} \sum_{t=1}^N \sum_{k=1}^K \sum_{j=1}^q \sum_{X_j \in \mathcal{X}_j} (-1)^{j+1} \left[ \left( \prod_{m=1}^M \left[ \boldsymbol{z}_{k,X_j,t}^{(m)} - l_k^{(m)} \right]_+ \right) \prod_{\boldsymbol{x}' \in X_j} \prod_{v=1}^V s(c^{(v)}(\boldsymbol{x}'); \epsilon) \right] \tag{12c}$$

### A.3.1 Inclusion Exclusion principle for $\text{HVI}_{\text{C}}$

Equation (10) holds when the indicator function because $\text{HVI}_{\text{C}}$ is equivalent to HVI with the subset of feasible points. However, the sigmoid approximation can result in non-zero error. The error function $\varepsilon : 2^{\mathcal{X}_{\text{cand}}} \to \mathbb{R}$ can be expressed as

$$\varepsilon(X) = \prod_{\boldsymbol{x}' \in X} \prod_{v=1}^V \mathbb{1}[c(\boldsymbol{x}') > 0] - \prod_{\boldsymbol{x}' \in X} \prod_{v=1}^V s(c(\boldsymbol{x}'), \epsilon)$$

The error function gives a value to each to each element of $2^{\mathcal{X}_{\text{cand}}}$. Weight functions have been studied in conjunction with the inclusion-exclusion principle [56], but under the assumption of that the weight of a set is the sum of the weights of its elements: $w(A) = \sum_{a \in A} w(a)$. In our case, the weight function of a set $A$ is the product the weights of its elements. There, it is not obvious whether the inclusion-exclusion principle will hold in this case.

**Theorem 1.** *Given a feasible Pareto front $\mathcal{P}_{\text{feas}}$, a partitioning $\{(\boldsymbol{l}_k, \boldsymbol{u}_k\}_{k=1}^K$ of the objective space $\mathbb{R}^M$ that is not dominated by the $\mathcal{P}_{\text{feas}}$, then for a set of points $\mathcal{X}_{\text{cand}}$ with objective values $\boldsymbol{f}(\mathcal{X}_{\text{cand}})$ and constraint values $\boldsymbol{c}(\mathcal{X}_{\text{cand}})$,*

$$\text{HVI}_{\text{C}}(\boldsymbol{f}(\mathcal{X}_{\text{cand}}), \boldsymbol{c}(\mathcal{X}_{\text{cand}}), \mathcal{P}, \boldsymbol{r}) = \text{HVI}(\boldsymbol{f}'(\mathcal{X}_{\text{cand}}), \mathcal{P}', \boldsymbol{r}')$$

*where $\boldsymbol{f}'(\mathcal{X}_{\text{cand}})$ is the set of objective-constraint vectors for each candidate point $\boldsymbol{f}'(\boldsymbol{x}) \in \mathbb{R}^{M+V}$, $\mathcal{P}'$ is the set of vectors $[f^{(1)}(\boldsymbol{x}), ..., f^{(M)}(\boldsymbol{x}), \boldsymbol{0}_V] \in \mathbb{R}^{M+V}$, and $\boldsymbol{r}' = [r^{(1)}, ..., r^{(M)}, \boldsymbol{0}_V] \in \mathbb{R}^{M+V}$.*

*Proof.* Recall equation 10

$$\text{HVI}_{\text{C}}(\{\boldsymbol{f}(\boldsymbol{x}_i), \boldsymbol{c}(\boldsymbol{x}_i)\}_{i=1}^q) = \sum_{k=1}^K \sum_{j=1}^q \sum_{X_j \in \mathcal{X}_j} (-1)^{j+1} \left[ \left( \prod_{m=1}^M \left[ z_{k,X_j}^{(m)} - l_k^{(m)} \right]_+ \right) \prod_{\boldsymbol{x}' \in X_j} \prod_{v=1}^V \mathbb{1}[c^{(v)}(\boldsymbol{x}') \geq 0] \right].$$

Note that the constraint product

$$\prod_{\boldsymbol{x}' \in X_j} \prod_{v=1}^{V} \mathbb{1}[c^{(v)}(\boldsymbol{x}') \geq 0] = \prod_{v=1}^{V} \prod_{\boldsymbol{x}' \in X_j} \mathbb{1}[c^{(v)}(\boldsymbol{x}') \geq 0]$$

$$= \prod_{v=1}^{V} \min_{\boldsymbol{x}' \in X_j} \mathbb{1}[c^{(v)}(\boldsymbol{x}') \geq 0]$$

$$= \prod_{v=1}^{V} \min\left[1, \min_{\boldsymbol{x}' \in X_j} \mathbb{1}[c^{(v)}(\boldsymbol{x}') \geq 0]\right] \quad (13)$$

$$= \prod_{v=1}^{V} \left[\min\left[1, \min_{\boldsymbol{x}' \in X_j} \mathbb{1}[c^{(v)}(\boldsymbol{x}') \geq 0]\right] - 0\right].$$

For $v = 1, \ldots, V$, $k = 1, \ldots K$, let $l_k^{(M+v)} = 0$ and $u_k^{(M+v)} = 1$. Then, substituting into the following expression from Equation 13 gives

$$\min\left[1, \min_{\boldsymbol{x}' \in X_j} \mathbb{1}[c^{(v)}(\boldsymbol{x}') \geq 0]\right] = \min\left[u_k^{(M+v)}, \min_{\boldsymbol{x}' \in X_j} \mathbb{1}[c^{(v)}(\boldsymbol{x}') \geq 0]\right]$$

Recall from Section 4, that $z$ is defined as: $\boldsymbol{z}_k := \min\left[\boldsymbol{u}_k, \boldsymbol{f}(\boldsymbol{x})\right]$. The high-level idea is that if we consider the indicator of the slack constraints $\mathbb{1}[c^{(v)}(\boldsymbol{x}') \geq 0]$ as objectives, then the above expression is consistent with the definition of $z$ at the beginning of section 4. For $v = 1, \ldots, V$,

$$z_{k,X_j}^{(M+v)} = \min\left[1, \min_{\boldsymbol{x}' \in X_j} \mathbb{1}[c^{(v)}(\boldsymbol{x}') \geq 0]\right]$$

Thus,

$$\prod_{\boldsymbol{x}' \in X_j} \prod_{v=1}^{V} \mathbb{1}[c^{(v)}(\boldsymbol{x}') \geq 0] = \prod_{v=1}^{V} \left[\min\left[1, \min_{\boldsymbol{x}' \in X_j} \mathbb{1}[c^{(v)}(\boldsymbol{x}') \geq 0]\right] - 0\right]$$

$$= \prod_{v=1}^{V} [z_{k,X_j}^{(M+v)} - l_k^{(M+v)}]_+$$

Returning to the $\text{HVI}_\text{C}$ equation, we have

$$\text{HVI}_\text{C}(\{\boldsymbol{f}(\boldsymbol{x}_i), \boldsymbol{c}(\boldsymbol{x}_i)\}_{i=1}^{q}) = \sum_{k=1}^{K}\sum_{j=1}^{q}\sum_{X_j \in \mathcal{X}_j} (-1)^{j+1}\left[\left(\prod_{m=1}^{M} [z_{k,X_j}^{(m)} - l_k^{(m)}]_+\right) \prod_{\boldsymbol{x}' \in X_j}\prod_{v=1}^{V} \mathbb{1}[c^{(v)}(\boldsymbol{x}') \geq 0]\right]$$

$$= \sum_{k=1}^{K}\sum_{j=1}^{q}\sum_{X_j \in \mathcal{X}_j} (-1)^{j+1}\left[\left(\prod_{m=1}^{M} [z_{k,X_j}^{(m)} - l_k^{(m)}]_+\right) \prod_{v=M+1}^{M+V} [z_{k,X_j}^{(v)} - l_k^{(M+v)}]_+\right]$$

$$= \sum_{k=1}^{K}\sum_{j=1}^{q}\sum_{X_j \in \mathcal{X}_j} (-1)^{j+1}\left[\prod_{m=1}^{M+V} [z_{k,X_j}^{(m)} - l_k^{(m)}]_+\right] \quad (14)$$

□

Now consider the case when a sigmoid approximation $\mathbb{1}[c^{(v)}(\boldsymbol{x}') \geq 0] \approx s(c^{(v)}(\boldsymbol{x}'); \epsilon)$ is used. The only change to Equation 14 is that

$$z_{k,X_j}^{(m)} \approx \hat{z}_{k,X_j}^{(m)} = \min\left[u_k^{(M+v)}, \min_{\boldsymbol{x}' \in X_j} S[c^{(v)}(\boldsymbol{x}'), \epsilon]\right].$$

If $S[c^{(v)}(\boldsymbol{x}'), \epsilon] = \mathbb{1}[c^{(v)}(\boldsymbol{x}') \geq 0]$ for all $v, \boldsymbol{x}'$, then HVI is computed exactly without approximation error. If $S[c^{(v)}(\boldsymbol{x}'), \epsilon]\mathbb{1}[c^{(v)}(\boldsymbol{x}') \geq 0]$ for any $v, \boldsymbol{x}'$, then there is approximation error: the hypervolume improvement from all subsets containing $\boldsymbol{x}'$ is proportional to $\prod_{v=1}^{V} \min_{\boldsymbol{x}' \in X} s(c(\boldsymbol{x}'), \epsilon)$. Since the constraint outcomes are directly considered as components in the hypervolume computation, the inclusion-exclusion principle incorporates the approximate indicator properly.

## A.4 Complexity

Recall from Section 3.3 that, given posterior samples, the time complexity on a single-threaded machine is $T_1 = O(MNK(2^q - 1))$. The space complexity required for maximum parallelism is also is $T_1$ (ignoring the space required by the models), which does limit scalability to larger $M$ and $q$, but difficulty scaling to large $M$ is a known limitaiton of EHVI [69]. To reduce memory load, rectangles could be materialized and processed in chunks at the cost of additional runtime. In addition, our implementation of qEHVI uses the box decomposition algorithm from Couckuyt et al. [11], but we emphasize qEHVI is agnostic to the choice of partitioning algorithm and using a more efficient partitioning algorithm (e.g. [69, 17, 41]) may significantly improve memory footprint on GPU and enable larger using $q$ in many scenarios.

# B  Error Bound on Sequential Greedy Approximation

If the acquisition function $\mathcal{L}(\mathcal{X}_{\text{cand}})$ is a normalized, monotone, submodular set function (where submodular means that the increase in $\mathcal{L}(\mathcal{X}_{\text{cand}})$ is non-increasing as elements are added to $\mathcal{X}_{\text{cand}}$ and normalized means that $\mathcal{L}(\emptyset) = 0$), then the sequential greedy approximation of $\mathcal{L}$ enjoys regret of no more than $\frac{1}{e}\mathcal{L}^*$, where $\mathcal{L}^*$ is the optima of $\mathcal{L}$ [23]. We have $\alpha_{q\text{EHVI}}(\mathcal{X}_{\text{cand}}) = \mathcal{L}(\mathcal{X}_{\text{cand}}) = \mathbb{E}_{\boldsymbol{f}}\big(\text{HVI}\big[\boldsymbol{f}(\mathcal{X}_{\text{cand}})\big]\big)$. Since HVI is a submodular set function [24] and the expectation of a stochastic submodular function is also submodular [2], $\alpha_{q\text{EHVI}}(\mathcal{X}_{\text{cand}})$ is also submodular and therefore its sequential greedy approximation enjoys regret of no more than $\frac{1}{e}\alpha_{q\text{EHVI}}^*$. Using the result from Wilson et al. [65], the MC-based approximation $\hat{\alpha}_{q\text{EHVI}}(\mathcal{X}_{\text{cand}}) = \sum_{t=1}^N \text{HVI}\big[\boldsymbol{f}_t(\mathcal{X}_{\text{cand}})\big]$ also enjoys the same regret bound since HVI is a normalized submodular set function.[7]

# C  Convergence Results

For the purpose of stating our convergence results, we recall some concepts and notation from Balandat et al. [5]. First, consider a sample $\{\boldsymbol{f}_t(\boldsymbol{x}_1)\}_{i=1}^q$ from the multi-output posterior of the GP surrogate model. Let $\boldsymbol{x} \in \mathbb{R}^{qd}$ be the stacked set of candidates $\mathcal{X}_{\text{cand}}$ and let $\boldsymbol{f}_t(\boldsymbol{x}) := [f_t(\boldsymbol{x}_1)^T, \ldots, f_t(\boldsymbol{x}_q)^T]^T$ be the stacked set of corresponding objective vectors. It is well known that, using the reparameterization trick, we can write

$$\boldsymbol{f}_t(\boldsymbol{x}) = \mu(\boldsymbol{x}) + L(\boldsymbol{x})\epsilon_t, \tag{15}$$

where $\mu : \mathbb{R}^{qd} \to \mathbb{R}^{qM}$ is the mean function of the multi-output GP, $L(\boldsymbol{x}) \in \mathbb{R}^{qM \times qM}$ is a root decomposition (typically the Cholesky decomposition) of the multi-output GP's posterior covariance $\Sigma(\boldsymbol{x}) \in \mathbb{R}^{qM \times qM}$, and $\epsilon_t \in \mathbb{R}^{qM}$ with $\epsilon_t \sim \mathcal{N}(0, I_{qM})$.

For $\boldsymbol{x} \in \mathcal{X}$, consider the MC-approximation $\hat{\alpha}_{q\text{EHVI}}^N(\boldsymbol{x})$ from (5). Denote by $\nabla_{\boldsymbol{x}}\hat{\alpha}_{q\text{EHVI}}^N(\boldsymbol{x})$ the gradient of $\hat{\alpha}_{q\text{EHVI}}^N(\boldsymbol{x})$, obtained by averaging the gradients on the sample-level:

$$\nabla_{\boldsymbol{x}}\hat{\alpha}_{q\text{EHVI}}^N(\boldsymbol{x}) := \frac{1}{N}\sum_{t=1}^N \nabla_{\boldsymbol{x}}\text{HVI}(\{f_t(\boldsymbol{x}_i)\}_{i=1}^q) \tag{16}$$

Let $\alpha_{q\text{EHVI}}^* := \max_{\boldsymbol{x} \in \mathcal{X}} \alpha_{q\text{EHVI}}(\boldsymbol{x})$ denote the maximum of the true acquisition function qEHVI, and let $\mathcal{X}^* := \arg\max_{\boldsymbol{x} \in \mathcal{X}} \alpha_{q\text{EHVI}}(\boldsymbol{x})$ denote the set of associated maximizers.

**Theorem 2.** *Suppose that $\mathcal{X}$ is compact and that $f$ has a Multi-Output Gaussian Process prior with continuously differentiable mean and covariance functions. If the base samples $\{\epsilon_t\}_{t=1}^N$ are drawn i.i.d. from $\mathcal{N}(0, I_{qM})$, and if $\hat{\boldsymbol{x}}_N^* \in \arg\max_{\boldsymbol{x} \in \mathcal{X}} \hat{\alpha}_{q\text{EHVI}}^N(\boldsymbol{x})$, then*

*(1) $\alpha_{q\text{EHVI}}(\hat{\boldsymbol{x}}_N^*) \to \alpha_{q\text{EHVI}}^*$ a.s.*

*(2) $\text{dist}(\hat{\boldsymbol{x}}_N^*, \mathcal{X}^*) \to 0$ a.s.*

In addition to the almost sure convergence in Theorem 2, deriving a result on the convergence rate of the optimizer, similar to the one obtained in [5], should be possible. We leave this to future work. Moreover, the results in Theorem 2 can also be extended to the situation in which the base samples are generated using a particular class of randomized QMC methods (see similar results in [5]).

*Proof.* We consider the setting from Balandat et al. [5, Section D.5]. Let $\epsilon \sim \mathcal{N}(0, I_{qM})$, so that we can write the posterior over outcome $m$ at $\boldsymbol{x}$ as the random variable $f^{(m)}(\boldsymbol{x}, \epsilon) = S_{\{i_j, m\}}(\mu(\boldsymbol{x}) + L(\boldsymbol{x})\epsilon)$, where $\mu(\boldsymbol{x})$

and $L(\boldsymbol{x})$ are the (vector-valued) posterior mean and the Cholesky factor of posterior covariance, respectively, and $S_{\{i_j,m\}}$ is an appropriate selection matrix (in particular, $\|S_{\{i_j,m\}}\|_\infty \leq 1$ for all $i_j$ and $m$). Let

$$A(\boldsymbol{x}, \epsilon) = \sum_{k=1}^{K} \sum_{j=1}^{q} \sum_{X_j \in \mathcal{X}_j} (-1)^{j+1} \prod_{m=1}^{M} \left[ z_{k,X_j}^{(m)}(\epsilon) - l_k^{(m)} \right]_+$$

where

$$z_{k,X_j}^{(m)}(\epsilon) = \min \left[ u_k^{(m)}, f^{(m)}(\boldsymbol{x}_{i_1}, \epsilon), \dots, f^{(m)}(\boldsymbol{x}_{i_j}, \epsilon) \right]$$

and $X_j = \{\boldsymbol{x}_{i_1}, \dots, \boldsymbol{x}_{i_j}\}$. Following [5, Theorem 3], we need to show that there exists an integrable function $\ell : \mathbb{R}^{q \times M} \mapsto \mathbb{R}$ such that for almost every $\epsilon$ and all $\boldsymbol{x}, \boldsymbol{y} \subseteq \mathcal{X}, \boldsymbol{x}, \boldsymbol{y} \in \mathbb{R}^{q \times d}$,

$$|A(\boldsymbol{x}, \epsilon) - A(\boldsymbol{y}, \epsilon)| \leq \ell(\epsilon)\|\boldsymbol{x} - \boldsymbol{y}\|. \tag{17}$$

Let us define

$$\tilde{a}_{kmjX_j}(\boldsymbol{x}, \epsilon) := \left[ \min \left[ u_k^{(m)}, f^{(m)}(\boldsymbol{x}_{i_1}, \epsilon), \dots, f^{(m)}(\boldsymbol{x}_{i_j}, \epsilon) \right] - l_k^{(m)} \right]_+.$$

Linearity implies that it suffices to show that this condition holds for

$$\tilde{A}(\boldsymbol{x}, \epsilon) := \prod_{m=1}^{M} \tilde{a}_{kmjX_j}(\boldsymbol{x}, \epsilon) = \prod_{m=1}^{M} \left[ \min \left[ u_k^{(m)}, f^{(m)}(\boldsymbol{x}_{i_1}, \epsilon), \dots, f^{(m)}(\boldsymbol{x}_{i_j}, \epsilon) \right] - l_k^{(m)} \right]_+ \tag{18}$$

for all $k, j$, and $X_j$. Observe that

$$\tilde{a}_{kmjX_j}(\boldsymbol{x}, \epsilon) \leq \left| \min \left[ u_k^{(m)}, f^{(m)}(\boldsymbol{x}_{i_1}, \epsilon), \dots, f^{(m)}(\boldsymbol{x}_{i_j}, \epsilon) \right] - l_k^{(m)} \right|$$

$$\leq |l_k^{(m)}| + \left| \min \left[ u_k^{(m)}, f^{(m)}(\boldsymbol{x}_{i_1}, \epsilon), \dots, f^{(m)}(\boldsymbol{x}_{i_j}, \epsilon) \right] \right|.$$

Note that if $u_k^{(m)} = \infty$, then $\min[u_k^{(m)}, f(\boldsymbol{x}, \epsilon)_{i_1}^{(m)}, \dots f^{(m)}(\boldsymbol{x}_{i_j}, \epsilon)] = \min[f^{(m)}(\boldsymbol{x}_{i_1}, \epsilon), \dots f^{(m)}(\boldsymbol{x}_{i_j}, \epsilon)]$. If $u_k^{(m)} < \infty$, then $\min[u_k^{(m)}, f^{(m)}(\boldsymbol{x}_{i_1}, \epsilon), \dots f^{(m)}(\boldsymbol{x}_{i_j}, \epsilon)] < \left| \min[f^{(m)}(\boldsymbol{x}_{i_1}, \epsilon), \dots f^{(m)}(\boldsymbol{x}_{i_j}, \epsilon)] \right| + |u_k^{(m)}|$. Let $w_k^{(m)} = u_k^{(m)}$ if $u_k^{(m)} < \infty$ and 0 otherwise. Then

$$\tilde{a}_{kmjX_j}(\boldsymbol{x}, \epsilon) \leq |l_k^{(m)}| + |w_k^{(m)}| + \left| \min \left[ f^{(m)}(\boldsymbol{x}_{i_1}, \epsilon), \dots, f^{(m)}(\boldsymbol{x}_{i_j}, \epsilon) \right] \right|$$

$$\leq |l_k^{(m)}| + |w_k^{(m)}| + \sum_{i_1, \dots, i_j} \left| f^{(m)}(\boldsymbol{x}_{i_j}, \epsilon) \right|.$$

We therefore have that

$$\left| \tilde{a}_{kmjX_j}(\boldsymbol{x}, \epsilon) \right| \leq |l_k^{(m)}| + |w_k^{(m)}| + |X_j| \left( \|\mu^{(m)}(\boldsymbol{x})\| + \|L^{(m)}(\boldsymbol{x})\| \|\epsilon\| \right)$$

for all $k, m, j, X_j$, where $|X_j|$ denotes the cardinality of the set $X_j$. Under our assumptions (compactness of $\mathcal{X}$, continuous differentiability of mean and covariance function), both $\mu(\boldsymbol{x})$ and $L(\boldsymbol{x})$, as well as their respective gradients w.r.t. $\boldsymbol{x}$, are uniformly bounded. In particular there exist $C_1, C_2 < \infty$ such that

$$\left| \tilde{a}_{kmjX_j}(\boldsymbol{x}, \epsilon) \right| \leq C_1 + C_2 \|\epsilon\|$$

for all $k, m, j, X_j$.

Dropping indices $k, j, X_j$ for simplicity, observe that

$$\left| \tilde{A}(\boldsymbol{x}, \epsilon) - \tilde{A}(\boldsymbol{y}, \epsilon) \right| = \left| \tilde{a}_1(\boldsymbol{x}, \epsilon)\tilde{a}_2(\boldsymbol{x}, \epsilon) - \tilde{a}_1(\boldsymbol{y}, \epsilon)\tilde{a}_2(\boldsymbol{y}, \epsilon) \right| \tag{19a}$$

$$= \left| \tilde{a}_1(\boldsymbol{x}, \epsilon)\left( \tilde{a}_2(\boldsymbol{x}, \epsilon) - \tilde{a}_2(\boldsymbol{y}, \epsilon) \right) + \tilde{a}_2(\boldsymbol{y}, \epsilon)\left( \tilde{a}_1(\boldsymbol{x}, \epsilon) - \tilde{a}_1(\boldsymbol{y}, \epsilon) \right) \right| \tag{19b}$$

$$\leq \left| \tilde{a}_1(\boldsymbol{x}, \epsilon) \right| \left| \tilde{a}_2(\boldsymbol{x}, \epsilon) - \tilde{a}_2(\boldsymbol{y}, \epsilon) \right| + \left| \tilde{a}_2(\boldsymbol{y}, \epsilon) \right| \left| \tilde{a}_1(\boldsymbol{x}, \epsilon) - \tilde{a}_1(\boldsymbol{y}, \epsilon) \right|. \tag{19c}$$

Furthermore,

$$\left| \tilde{a}_{kmjX_j}(\boldsymbol{x}, \epsilon) - \tilde{a}_{kmjX_j}(\boldsymbol{y}, \epsilon) \right| \leq \sum_{i_1, \dots, i_j} \left| S_{\{i_j,m\}}(\mu(\boldsymbol{x}) + L(\boldsymbol{x})\epsilon) - S_{\{i_j,m\}}(\mu(\boldsymbol{y}) + L(\boldsymbol{y})\epsilon) \right|$$

$$\leq |X_j| \left( \|\mu(\boldsymbol{x}) - \mu(\boldsymbol{y})\| + \|L(\boldsymbol{x}) - L(\boldsymbol{y})\| \|\epsilon\| \right).$$

Since $\mu$ and $L$ have uniformly bounded gradients, they are Lipschitz. Therefore, there exist $C_3, C_4 < \infty$ such that

$$\left| \tilde{a}_{kmjX_j}(\boldsymbol{x}, \epsilon) - \tilde{a}_{kmjX_j}(\boldsymbol{y}, \epsilon) \right| \leq (C_3 + C_4 \|\epsilon\|) \|\boldsymbol{x} - \boldsymbol{y}\|$$

for all $\boldsymbol{x}, \boldsymbol{y}, k, m, j, X_j$. Plugging this into (19) above, we find that

$$\left| \tilde{A}(\boldsymbol{x}, \epsilon) - \tilde{A}(\boldsymbol{y}, \epsilon) \right| \leq 2 \Big( C_1 C_3 + (C_1 C_4 + C_2 C_3) \|\epsilon\| + C_2 C_4 \|\epsilon\|^2 \Big) \|\boldsymbol{x} - \boldsymbol{y}\|$$

for all $\boldsymbol{x}, \boldsymbol{y}$ and $\epsilon$. For $M > 2$ we generalize the idea from (19), making sure to telescope the respective expressions. It is not hard to see that with this, there exist $C < \infty$ such that

$$\left| \tilde{A}(\boldsymbol{x}, \epsilon) - \tilde{A}(\boldsymbol{y}, \epsilon) \right| \leq C \sum_{m=1}^{M} \|\epsilon\|^m \|\boldsymbol{x} - \boldsymbol{y}\|$$

Letting $\ell(\epsilon) := C \sum_{m=1}^{M} \|\epsilon\|^m$, we observe that $\ell(\epsilon)$ is integrable (since all absolute moments exist for the Normal distribution).

The result now follows from in Balandat et al. [5, Theorem 3]. $\qquad \square$

Besides the above convergence result, we can also show that the sample average gradient of the MC approximation of $q$EHVI is an unbiased estimator of the true gradient of $q$EHVI:

**Proposition 1.** *Suppose that the GP mean and covariance function are continuously differentiable. Suppose further that the candidate set $\boldsymbol{x}$ has no duplicates, and that the sample-level gradients $\nabla_{\boldsymbol{x}} \mathrm{HVI}(\{f_t(\boldsymbol{x}_i)\}_{i=1}^q)$ are obtained using the reparameterization trick as in [5]. Then*

$$\mathbb{E}\left[ \nabla_{\boldsymbol{x}} \hat{\alpha}_{q\mathrm{EHVI}}^N(\boldsymbol{x}) \right] = \nabla_{\boldsymbol{x}} \alpha_{q\mathrm{EHVI}}(\boldsymbol{x}), \tag{20}$$

*that is, the averaged sample-level gradient is an unbiased estimate of the gradient of the true acquisition function.*

*Proof.* This proof follows the arguments Wang et al. [63, Theorem 1], which leverages Glasserman [31, Theorem 1]. We verify the conditions of Glasserman [31, Theorem 1] below. Using the arguments from [5], we know that, under the assumption of differentiable mean and covariance functions, the samples $\boldsymbol{f}_t(\boldsymbol{x})$ are continuously differentiable w.r.t. $\boldsymbol{x}$ (since there are no duplicates, and thus the covariance $\Sigma(\boldsymbol{x})$ is non-singular). Hence, Glasserman [31, A1] is satisfied. Furthermore, it is easy to see from (1) that $\mathrm{HVI}(\{\boldsymbol{f}(\boldsymbol{x}_i)\}_{i=1}^q)$ is *a.s.* continuous and is differentiable w.r.t. $\boldsymbol{f}_t(\boldsymbol{x})$ on $\mathbb{R}^M$, except on the edges of the hyper-rectangle decomposition $\{S_k\}_{k=1}^K$ of the non-dominated space, which satisfies [31, A3]. The set of points defined by the union of these edges clearly has measure zero under any non-degenerate (non-singular covariance) GP posterior on $\mathbb{R}^M$, so Glasserman [31, A4] holds. Therefore Glasserman [31, Lemma 2] holds, so $\mathrm{HVI}(\{\boldsymbol{f}(\boldsymbol{x}_i)\}_{i=1}^q)$ is *a.s.* piece-wise differentiable w.r.t. $\boldsymbol{x}$.

Lastly, we need to show that the result in Glasserman [31, Lemma 3] holds:

$$\mathbb{E}\left[ \sup_{x_{ci} \notin \tilde{D}} |A'(\boldsymbol{x}, \epsilon)| \right] < \infty.$$

As in Wang et al. [63, Theorem 1], we fix $\boldsymbol{x}$ except for $x_{ci}$ where $x_{ci}$ is the $c^{\text{th}}$ component of the $i^{\text{th}}$ point, We need to show that $\mathbb{E}\left[ \sup_{x_{ci} \notin \tilde{D}} |A'(\boldsymbol{x}, \epsilon)| \right] < \infty$. By linearity, it suffices to show that $\mathbb{E}\left[ \sup_{x_{ci} \notin \tilde{D}} |\tilde{A}'(\boldsymbol{x}, \epsilon)| \right] < \infty$. We have

$$\mathbb{E}\left[ \sup_{x_{ci} \notin \tilde{D}} |\tilde{A}'(\boldsymbol{x}, \epsilon)| \right] = \mathbb{E}\left[ \sup_{x_{ci} \notin \tilde{D}} \left| \frac{\partial \tilde{A}(\boldsymbol{x}, \epsilon)}{\partial x_{ci}} \right| \right].$$

Consider the $M = 2$ case. We have $\tilde{A}(\boldsymbol{x}, \epsilon) = a_1(\boldsymbol{x}, \epsilon) a_2(\boldsymbol{x}, \epsilon)$, where

$$a_m(\boldsymbol{x}, \epsilon) = \left[ \min \left[ u_k^{(m)}, f^{(m)}(\boldsymbol{x}_{i_1}, \epsilon), \dots, f^{(m)}(\boldsymbol{x}_{i_j}, \epsilon) \right] - l_k^{(m)} \right]_+.$$

The partial derivative of $\tilde{A}(\boldsymbol{x}, \epsilon)$ with respect to $x_{ci}$ is

$$\frac{\partial \tilde{A}(\boldsymbol{x}, \epsilon)}{\partial x_{ci}} = \frac{\partial a_1(\boldsymbol{x}, \epsilon)}{\partial x_{ci}} a_2(\boldsymbol{x}, \epsilon) + a_1(\boldsymbol{x}, \epsilon) \frac{\partial a_2(\boldsymbol{x}, \epsilon)}{\partial x_{ci}},$$

and therefore

$$\left| \frac{\partial \tilde{A}(\boldsymbol{x}, \epsilon)}{\partial x_{ci}} \right| \leq \left| \frac{\partial a_1(\boldsymbol{x}, \epsilon)}{\partial x_{ci}} \right| \cdot \left| a_2(\boldsymbol{x}, \epsilon) \right| + \left| a_1(\boldsymbol{x}, \epsilon) \right| \cdot \left| \frac{\partial a_2(\boldsymbol{x}, \epsilon)}{\partial x_{ci}} \right|$$

Since we are only concerned with $x_{ci} \notin \tilde{D}$,

$$a_m(\boldsymbol{x}, \epsilon) = \left[ \min \left[ f^{(m)}(\boldsymbol{x}_{i_1}, \epsilon), \dots, f^{(m)}(\boldsymbol{x}_{i_j}, \epsilon) \right] - l_k^{(1)} \right]_+.$$

As in the proof of Theorem 2, we write the posterior over outcome $m$ at $\boldsymbol{x}$ as the random variable $f^{(m)}(\boldsymbol{x}, \epsilon) = S_{\{i_j, m\}}(\boldsymbol{\mu}(\boldsymbol{x}) + L(\boldsymbol{x})\epsilon)$, where $\epsilon \sim \mathcal{N}(0, I_{qM})$ and $S_{\{i_j, m\}}$ is an appropriate selection matrix. With this,

$$a_m(\boldsymbol{x}, \epsilon) = \left[\min\left[S_{\{i_1, 1\}}(\mu(\boldsymbol{x}) + L(\boldsymbol{x})\epsilon), \dots, S_{\{i_j, 1\}}(\mu(\boldsymbol{x}) + L(\boldsymbol{x})\epsilon)\right] - l_k^{(1)}\right]_+.$$

Since the interval $\mathcal{X}$ is compact and the mean, covariance, and Cholesky factor of the covariance $\mu(\boldsymbol{x}), C(\boldsymbol{x}), L(\boldsymbol{x})$ are continuously differentiable, for all $m$ we have

$$\sup_{x_{ci}}\left|\frac{\partial \mu^{(m)}(\boldsymbol{x}_a)}{\partial x_{ci}}\right| = \mu_a^{*,(m)} < \infty, \qquad \sup_{x_{ci}}\left|\frac{\partial L^{(m)}(\boldsymbol{x})}{\partial x_{ci}}\right| = L_{ca}^{*,(m)} < \infty.$$

Let $\mu_{**}^{(m)} = \max_a \mu_a^{*,(m)}$, $L_{**}^{(m)} = \max_{a,b} L_{ab}^{*,(m)}(\boldsymbol{x})$, where $L_{ab}^{(m)}$ is the element at row $a$, column $b$ in $L^{(m)}$, the Cholesky factor for outcome $m$. Let $\epsilon^{(m)} \in \mathbb{R}^q$ denote the vector of i.i.d. $\mathcal{N}(0,1)$ samples corresponding to outcome $m$. Then we have

$$\left|\frac{\partial}{\partial x_{ci}}\left[\left[\min\left[S_{\{i_1, 1\}}(\mu(\boldsymbol{x}) + L(\boldsymbol{x})\epsilon), \dots, S_{\{i_j, 1\}}(\mu(\boldsymbol{x}) + L(\boldsymbol{x})\epsilon)\right] - l_k^{(1)}\right]_+\right]\right|$$

$$\leq \left|\left[\mu_{**}^{(m)} + L_{**}^{(m)}||\epsilon^{(m)}||_1 - l_k^{(m)}\right]_+\right|$$

$$\leq \left|\mu_{**}^{(m)} + L_{**}^{(m)}||\epsilon^{(m)}||_1\right| + \left|l_k^{(m)}\right|.$$

Under our assumptions (compactness of $\mathcal{X}$, continuous differentiability of mean and covariance function) both $\boldsymbol{\mu}(\boldsymbol{x})$ and $L(\boldsymbol{x})$, as well as their respective gradients, are uniformly bounded. In particular there exist $C_1^{(m)}, C_2^{(m)} < \infty$ such that

$$\left|S_{\{a, m\}}(\mu(\boldsymbol{x}) + L(\boldsymbol{x})\epsilon) - l_k^{(m)}\right| \leq C_1^{(m)} + C_2^{(m)}||\epsilon^{(m)}||_1$$

for all $a = i_1, ..., i_j$.

Hence,

$$\left|\frac{\partial \tilde{A}(\boldsymbol{x}, \epsilon)}{\partial x_{ci}}\right| \leq \left[\left|\mu_{**}^{(1)} + C_{**}^{(1)}||\epsilon^{(1)}||_1\right| + \left|l_k^{(1)}\right|\right]\left[C_1^{(2)} + C_2^{(2)}||\epsilon^{(2)}||_1\right]$$

$$+ \left[C_1^{(1)} + C_2^{(1)}||\epsilon^{(1)}||_1\right]\left[\left|\mu_{**}^{(2)} + C_{**}^{(2)}||\epsilon^{(2)}||_1\right| + \left|l_k^{(2)}\right|\right]$$

Since $\epsilon$ is absolutely integrable,

$$\mathbb{E}\left(\left|\frac{\partial \tilde{A}(\boldsymbol{x}, \epsilon)}{\partial x_{ci}}\right|\right) < \infty.$$

Hence, $\mathbb{E}\left[\sup_{x_{ci} \notin \tilde{D}} |A'(\boldsymbol{x}, \epsilon)|\right] < \infty$. This can be extended to $M > 2$ in the same manner using the product rule to obtain

$$\mathbb{E}\left(\frac{\partial \tilde{A}(\boldsymbol{x}, \epsilon)}{\partial x_{ci}}\right) \leq \sum_{m=1}^{M}\left(\left[\left|\mu_{**}^{(m)} + C_{**}^{(m)}\mathbb{E}[||\epsilon^{(m)}||_1]\right| + \left|l_k^{(1)}\right|\right]\prod_{n=1, n\neq m}^{M}\left[C_1^{(n)} + C_2^{(n)}\mathbb{E}[||\epsilon^{(n)}||_1]\right]\right)$$

$$\leq \sum_{m=1}^{M}\left(\left[\left|\mu_{**}^{(m)} + \frac{\pi}{2}qC_{**}^{(m)}\right| + \left|l_k^{(1)}\right|\right]\prod_{n=1, n\neq m}^{M}\left[C_1^{(n)} + \frac{\pi}{2}qC_2^{(n)}\right]\right).$$

Hence, $\mathbb{E}\left[\sup_{x_{ci} \notin \tilde{D}} |A'(\boldsymbol{x}, \epsilon)|\right] < \infty$ for $M \geq 2$ and Glasserman [31, Theorem 1] holds. $\square$

# D Monte-Carlo Approximation

Figure 5b shows the gradient of analytic EHVI and the MC estimator $q$EHVI on slice of a 3-objective problem. Even using only $N = 32$ QMC samples, the average sample gradient has very low variance. Moreover, fixing the base samples also greatly reduces the variance without introducing bias.

(a) A comparison of the analytic EHVI acquisition function and the MC-based $q$EHVI for $q = 1$.

(b) A comparison of the exact gradient of analytic EHVI and the exact sample average gradient of the MC-based $q$EHVI for $q = 1$.

Figure 5: A comparison of (a) the analytic EHVI and MC-based $q$EHVI for $q = 1$ and (b) a comparison of the exact gradient $\nabla\alpha_{\text{EHVI}}$ of analytic EHVI and average sample gradient of the MC-estimator $\nabla\hat{\alpha}_{q\text{EHVI}}$ over a slice of the input space on a DTLZ2 problem ($q = 1$, $M = 3$, $d = 6$) [15]. $x^{(0)}$ is varied across $0 \le \lambda \le 1$, while $x^{(i)}$ for $1, ...D$ are held constant. In each of (a) and (b), the top row show $q$EHVI where the (quasi-)standard normal base samples are resampled for each value of $x^{(0)}$. The solid line is one sample average (across (q)MC samples) and the shaded area is the mean plus 2 standard errors across 50 repetitions. The bottom row uses the same base samples for evaluating each test point and the sample average for each of 50 repetitions is plotted.

# E Experiment Details

## E.1 Algorithms

For TS-TCH, we draw a sample from the joint posterior over a discrete set of $1000d$ points sampled from a scrambled Sobol sequence. For PESMO, we follow [27] and use a Pareto set of size 10 for each sampled GP, which is optimized over a discrete set of $1000d$ points sampled from a scrambled Sobol sequence. The current

Table 2: Reference points for all benchmark problems. Assuming minimization. In our benchmarks, equivalently maximize the negative objectives and multiply the reference points by -1.

| PROBLEM | REFERENCE POINT |
|---|---|
| BRANINCURRIN | (18.0, 6.0) |
| DTLZ2 | $(1.1, ..., 1.1) \in \mathbb{R}^M$ |
| ABR | (-150.0, 3500.0, 5.1) |
| VEHICLE CRASH SAFETY | (1864.72022, 11.81993945, 0.2903999384) |
| CONSTRAINEDBRANINCURRIN | (90.0, 10.0) |
| C2-DTLZ2 | $(1.1, ..., 1.1) \in \mathbb{R}^M$ |

Pareto front is approximated by optimizing the posterior means over a grid as is done in Garrido-Merchán and Hernández-Lobato [26, 27]. For SMS-EGO, we use the observed Pareto front. All acquisition functions are optimized with L-BFGS-B (with a maximum of 200 iterations); SMS-EGO [53] and PESMO [26] use gradients approximated by finite differences and all other methods use exact gradients. For all methods, each outcome is modeled with an independent Gaussian process with a Matern $5/2$ ARD kernel. The methods implemented in Spearmint use a fully Bayesian treatment of the hyperparameters with 10 samples from posterior over the hyperparamters, and the methods implemented in BoTorch use maximum a posteriori estimates of the GP hyperparameters. All methods are initialized with $2(d+1)$ points from a scrambled Sobol sequence. $q$PAREGO and $q$EHVI use $N = 128$ QMC samples.

### E.1.1 Reference point specification

There is a large body of literature on the effects of reference point specification [4, 35, 36]. The hypervolume indicator is sensitive to specified the reference point: a reference point that is far away from the Pareto front will favor extreme points, where as reference point that is close to the Pareto front gives more weight to less extreme points [36]. Sensitivity to the reference point is affects both the evaluation of different MO methods and the utility function for methods that rely HV. In practice, a decision maker may be able to specify a reference point that satisfies their preference with domain knowledge. If a reference point is provided by the decision maker, previous work has suggested heuristics for choosing reference points for use in an algorithm's utility function [35, 53]. We follow previous work [69, 68] and assume that the reference point is known.

We also considered (but did not use in our experiments) a dynamic reference point strategy where at each BO iteration, the reference point is selected to be a point slightly worse than the nadir (component-wise minimum) point of the current observed Pareto front for computing the acquisition function: $\boldsymbol{r} = \boldsymbol{y}_{\text{nadir}} - 0.1 \cdot |\boldsymbol{y}_{\text{nadir}}|$ where $\boldsymbol{y}_{\text{nadir}} = \left( \min_{y^{(1)} \in \mathcal{D}^{(1)}} y^{(1)}, \ldots, \min_{y^{(m)} \in \mathcal{D}^{(m)}} y^{(m)} \right)$. This reference point is used in SMS-EMOA in Ishibuchi et al. [35]), and we find similar average performance (but higher variance) on problems to using a known reference point with continuous Pareto fronts. If the Pareto front is discontinuous, then it is possible not all sections of the Pareto front will be reached.

### E.1.2 $q$PAREGO

Previous work has only considered unconstrained sequential optimization with ParEGO [40, 7] and ParEGO is often optimized with gradient-free methods [53]. To the best of our knowledge, $q$PAREGO is the first to support parallel and constrained optimization. Moreover, we compute exact gradients via auto-differentiation for acquisition optimization. ParEGO is typically implemented by applying augmented Chebyshev scalarization and modeling the scalarized outcome [40]. However, recent work has shown that composite objectives offer improved optimization performance [3]. $q$PAREGO uses a MC-based Expected Improvement [38] acquisition function, where the objectives are modeled independently and the augmented Chebyshev scalarization [40] is applied to the posterior samples as a composite objective. This approach enables the use of sequential greedy optimization of $q$ candidates with proper integration over the posterior at the pending points. Importantly, the sequential greedy approach allows for using different random scalarization weights for selecting each of the $q$ candidates. $q$PAREGO is extended to the constrained setting by weighting the EI by the probability of feasibility [25]. We estimate the probability of feasiblity using the posterior samples and approximate the indicator function with a sigmoid to maintain differentiablity as in constrained $q$EHVI. $q$PAREGO is trivially extended to the noisy setting using Noisy Expected Improvement [43, 5], but we use Expected Improvement in our experiments as all of the problems are noiseless.

### E.2 Benchmark Problems

The details for the benchmark problems below assume minimization of all objectives. Table 2 provides the reference points used for all benchmark problems.

**Branin-Currin**

$$f^{(1)}(x'_1, x'_2) = (x_2 - \frac{5.1}{4\pi^2}x_1^2 + \frac{5}{\pi}x_1 - r)^2 + 10(1 - \frac{1}{8\pi})\cos(x_1) + 10$$

$$f^{(2)}(x_1, x_2) = \left[1 - \exp\left(-\frac{1}{(2x_2)}\right)\right]\frac{2300x_1^3 + 1900x_1^2 + 2092x_1 + 60}{100x_1^3 + 500x_1^2 + 4x_1 + 20}$$

where $x_1, x_2 \in [0, 1]$, $x'_1 = 15x_1 - 5$, and $x'_2 = 15x_2$.

The constrained Branin-Currin problem uses the following disk constraint from [29]:

$$c(x'_1, x'_2) = 50 - (x'_1 - 2.5)^2 - (x'_2 - 7.5)^2) \geq 0$$

**DTLZ2** The objectives are given by [15]:

$$f_1(\boldsymbol{x}) = (1 + g(\boldsymbol{x}_M))\cos\left(\frac{\pi}{2}x_1\right)\cdots\cos\left(\frac{\pi}{2}x_{M-2}\right)\cos\left(\frac{\pi}{2}x_{M-1}\right)$$

$$f_2(\boldsymbol{x}) = (1 + g(\boldsymbol{x}_M))\cos\left(\frac{\pi}{2}x_1\right)\cdots\cos\left(\frac{\pi}{2}x_{M-2}\right)\sin\left(\frac{\pi}{2}x_{M-1}\right)$$

$$f_3(\boldsymbol{x}) = (1 + g(\boldsymbol{x}_M))\cos\left(\frac{\pi}{2}x_1\right)\cdots\sin\left(\frac{\pi}{2}x_{M-2}\right)$$

$$\vdots$$

$$f_M(\boldsymbol{x}) = (1 + g(\boldsymbol{x}_M))\sin\left(\frac{\pi}{2}x_1\right)$$

where $g(\boldsymbol{x}) = \sum_{x_i \in \boldsymbol{x}_M}(x_i - 0.5)^2$, $\boldsymbol{x} \in [0, 1]^d$, and $\boldsymbol{x}_M$ represents the last $d - M + 1$ elements of $\boldsymbol{x}$.

The C2-DTLZ2 problem adds the following constraint [16]:

$$c(\boldsymbol{x}) = -\min\left[\min_{i=1}^{M}\left((f_i(\boldsymbol{x}) - 1)^2 + \sum_{j=1, j=i}^{M}(f_j^2 - r^2)\right), \left(\sum_{i=1}^{M}\left((f_i(\boldsymbol{x}) - \frac{1}{\sqrt{M}})^2 - r^2\right)\right)\right] \geq 0$$

**Vehicle Crash Safety** The objectives are given by [60]:

$$f_1(\boldsymbol{x}) = 1640.2823 + 2.3573285x_1 + 2.3220035x_2 + 4.5688768x_3 + 7.7213633x_4 + 4.4559504x_5$$

$$f_2(\boldsymbol{x}) = 6.5856 + 1.15x_1 - 1.0427x_2 + 0.9738x_3 + 0.8364x_4 - 0.3695x_1x_4 + 0.0861x_1x_5$$
$$+ 0.3628x_2x_4 + 0.1106x_1^2 - 0.3437x_3^2 + 0.1764x_4^2$$

$$f_3(\boldsymbol{x}) = -0.0551 + 0.0181x_1 + 0.1024x_2 + 0.0421x_3 - 0.0073x_1x_2 + 0.024x_2x_3 - 0.0118x_2x_4$$
$$- 0.0204x_3x_4 - 0.008x_3x_5 - 0.0241x_2^2 + 0.0109x_4^2$$

where $\boldsymbol{x} \in [1, 3]^5$.

**Policy Optimization for Adaptive Bitrate Control** The controller is given by: $a_t = x_0\hat{z}_{\text{bd},t} + x_2 z_{\text{bf},t} + x_3$, where $\hat{z}_{\text{bd},t} = \frac{\sum_{t_i < t} z_{\text{bd},t_i}\exp(-x_1 t_i)}{\sum_{t_i < t}\exp(-x_1 t_i)}$ is estimated bandwidth at time $t$ using an exponential moving average, $z_{\text{bf},t}$ is the buffer occupancy at time $t$, and $x_0, \ldots x_3$ are the parameters we seek to optimize. We evaluate each policy on a set of 400 videos, where the number of time steps (chunks) in each video stream trajectory depends on the size of the video.

Table 3: Acquisition Optimization wall time in seconds on a CPU (2x Intel Xeon E5-2680 v4 @ 2.40GHz) and on a GPU (Tesla V100-SXM2-16GB). The mean and two standard errors are reported. NA indicates that the algorithm does not support constraints.

| CPU | CONSTRAINEDBRANINCURRIN | DTLZ2 |
|---|---|---|
| PESMO ($q$=1) | NA | 278.53 ($\pm$25.66) |
| SMS-EGO ($q$=1) | NA | 104.26 ($\pm$7.66) |
| TS-TCH ($q$=1) | NA | 52.55 ($\pm$0.06) |
| $q$PAREGO ($q$=1) | 2.4 ($\pm$0.37) | 4.68 ($\pm$0.46) |
| EHVI ($q$=1) | NA | 3.58 ($\pm$0.28) |
| $q$EHVI ($q$=1) | 5.69 ($\pm$0.43) | 5.95 ($\pm$0.45) |
| **GPU** | CONSTRAINEDBRANINCURRIN | DTLZ2 |
| TS-TCH ($q$=1) | NA | 0.25 ($\pm$0.00) |
| TS-TCH ($q$=2) | NA | 0.27 ($\pm$0.00) |
| TS-TCH ($q$=4) | NA | 0.28 ($\pm$0.00) |
| TS-TCH ($q$=8) | NA | 0.32 ($\pm$0.01) |
| $q$PAREGO ($q$=1) | 3.52 ($\pm$0.34) | 9.04 ($\pm$0.93) |
| $q$PAREGO ($q$=2) | 6.0 ($\pm$0.56) | 14.23 ($\pm$1.55) |
| $q$PAREGO ($q$=4) | 12.07 ($\pm$0.98) | 40.5 ($\pm$3.21) |
| $q$PAREGO ($q$=8) | 33.1 ($\pm$3.32) | 84.15 ($\pm$6.9) |
| EHVI ($q$=1) | NA | 84.15 ($\pm$6.9) |
| $q$EHVI ($q$=1) | 5.61 ($\pm$0.17) | 10.21 ($\pm$0.58) |
| $q$EHVI ($q$=2) | 19.06 ($\pm$5.88) | 17.75 ($\pm$0.97) |
| $q$EHVI ($q$=4) | 29.26 ($\pm$2.01) | 40.41 ($\pm$2.78) |
| $q$EHVI ($q$=8) | 91.56 ($\pm$5.51) | 106.51 ($\pm$7.69) |

# F  Additional Empirical Results

## F.1  Additional Sequential Optimization Results

We include results for an additional synthetic benchmark: the **DTLZ2** problem from the MO literature [15] ($d = 6, M = 2$). Figure 6 shows that $q$EHVI outperforms all other baseline algorithms on the DTLZ2 in terms of sequential optimization performance with competitive wall times as shown in 3.

Figure 6: Optimization performance on the DTLZ2 synthetic function ($d = 6, M = 2$).

## F.2  Performance with Increasing Parallelism

Figure 7 shows that that the performance of $q$EHVI performance does not degrade substantially, whereas performance does degrade for $q$PAREGO and TS-TCH on some benchmark problems. We include results for all problems in Section 5 and Appendix F.1 as well as a **Constrained Branin-Currin** problem (which is described in Appendix E.2).

(a) VEHICLESAFETY

(b) VEHICLESAFETY

(c) C2DTLZ2

(d) C2DTLZ2

(e) BRANINCURRIN

(f) BRANINCURRIN

Figure 7: Optimization performance of parallel acquisition functions over *batch BO iterations* (left) and *function evaluations* (right) for benchmark problems in Section 5.

(a) CONSTRAINEDBRANINCURRIN

(b) CONSTRAINEDBRANINCURRIN

(c) DTLZ2 ($M = 2, d = 6$)

(d) DTLZ2 ($M = 2, d = 6$)

Figure 8: Optimization performance of parallel acquisition functions over *batch BO iterations* (left) and *function evaluations* (right) for additional benchmark problems.

## F.3 Noisy Observations

Although neither $q$EHVI nor any variant of expected hypervolume improvement (to our knowledge) directly account for noisy observations, noisy observations are a practical challenge. We empirically evaluate the performance of all algorithms on a Branin-Currin function where observations have additive, zero-mean, $iid$ Gaussian noise; the unknown standard deviation of the noise is set to be $1\%$ of the range of each objective. Fig 9 shows that $q$EHVI performs favorably in the presence of noise, besting all algorithms including Noisy $q$PAREGO ($q$NParego) (described in Appendix E.1.2), PESMO and TS-TCH, all of which account for noise.

Figure 9: Sequential optimization performance on a noisy Branin-Currin problem.

## F.4 Approximate Box Decompositions

EHVI becomes prohibitively computationally expensive in many scenarios with $\geq 4$ objectives because of the wall time of partitioning the non-dominated space into disjoint rectangles [11]. Therefore, in addition to providing an exact binary partitioning algorithm, Couckuyt et al. [11] propose an approximation that terminates the partitioning algorithm when the new additional set of hyper-rectangles in the partitioning has a total hypervolume of less than a predetermined fraction $\zeta$ of the hypervolume dominated by the Pareto front. While $q$EHVI is guaranteed to be exact when an exact partitioning of the non-dominated space is used, $q$EHVI is agnostic to the partitioning algorithm used and is compatible with more scalable approximate methods.

We evaluate the performance of $q$EHVI with approximation of various fidelities $\zeta$ on DTLZ2 problems with 3 and 4 objectives (with $d = 6$). $\zeta = 0$ corresponds to an exact partitioning and the approximation is monotonically worse as $\zeta$ increases. Larger values of $\zeta$ degrade optimization performance (Figure 10), but can result in substantial speedups (Table 4). Even with coarser levels of approximation, $q$EHVI() performs better than $q$PAREGO with respect to log hypervolume difference, while achieving wall time improvements of 2-7x compared to exact $q$EHVI.

Figure 10: Optimization performance on DTLZ2 problems ($d = 6$) with approximate partitioning using various approximation levels $\zeta$ for (a) $M = 3$ objectives and (b) $M = 4$ objectives.

| CPU | DTLZ2 ($M = 3$) | DTLZ2 ($M = 4$) |
|---|---|---|
| $q$PAREGO | 5.86 ($\pm$0.51) | 5.6 ($\pm$0.53) |
| $q$EHVI ($\zeta = 10^{-3}$) | 6.89 ($\pm$0.41) | 9.53 ($\pm$0.49) |
| $q$EHVI ($\zeta = 10^{-4}$) | 9.83 ($\pm$0.9) | 17.47 ($\pm$1.2) |
| $q$EHVI ($\zeta = 10^{-5}$) | 18.99 ($\pm$2.72) | 60.27 ($\pm$3.57) |
| $q$EHVI ($\zeta = 10^{-6}$) | 37.9 ($\pm$7.47) | 136.15 ($\pm$12.88) |
| $q$EHVI (EXACT) | 45.52 ($\pm$9.83) | 459.33 ($\pm$77.95) |

Table 4: Acquisition function optimization wall time with approximate hypervolume computation, in seconds on a CPU (2x Intel Xeon E5-2680 v4 @ 2.40GHz). The mean and two standard errors are reported.

## F.5 Acquisition Computation Time

Figure 11 show the acquisition computation time for different $M$ and $q$. The inflection points corresponds to available processor cores becoming saturated. For large $M$ an $q$ on the GPU, memory becomes an issue, but we discuss ways of mitigating the issue in Appendix A.4.

Figure 11: Acquisition computation time for different batch sizes $q$ and numbers of objectives $M$ (this excludes the time required to compute the acquisition function *given* box decomposition of the non-dominated space). This uses $N = 512$ MC samples, $d = 6$, $|\mathcal{P}| = 10$, and 20 training points. CPU time was measured on 2x Intel Xeon E5-2680 v4 @ 2.40GHz and GPU time was measured on a Tesla V100-SXM2-16GB GPU using 64-bit floating point precision. The mean and 2 standard errors over 1000 trials are reported.

## Footnotes

[7]As noted in Wilson et al. [65], submodularity technically requires the search space $\mathcal{X}$ to be finite, whereas in BO, it will typically be infinite. Wilson et al. [65] note that in similar scenarios, submodularity has been extended to infinite sets $\mathcal{X}$ (e.g. Srinivas et al. [58]).