[Reviews · NeurIPS 2020]

Review 1

Summary and Contributions: This work describes a method for batch multi-objective constrained Bayesian optimization. The proposed method is very simple and based on expected hyper-volume improvement. This quantity is computationally is intractable for batches of size larger or equal to 2. The authors use a Monte Carlo approximation of the required integrals. Furthermore, the approximation is computationally expensive with respect to the number of objective (exponential). The authors suggest the use of parallel architectures for the evaluation such as GPUs. An advantage of the proposed method is that it allows for exact gradient computation. The proposed method is validated on 2 synthetic an 2 real-world problems.

Strengths: - Good written paper. - Good empirical results shown. - Simple idea.

Weaknesses: - There is not that much novelty in the proposed approach. - Only a few experiments are carried out. - The proposed approach is limited to noiseless settings. - The constrained setting could be problematic (see blow).

Correctness: I believe that the results shown and the methodology employed by the authors is correct.

Clarity: The paper is easy to read and clearly explained.

Relation to Prior Work: Related prior work is correctly cited and described.

Reproducibility: Yes

Additional Feedback: This is a nice paper that may attract the attention of the community working on Bayesian optimization. My main concern is that the proposed method is not that much novel. It simply consists in using an already known acquisition function, expected hyper-volume improvement, which is adapted for quick evaluation in a parallel setting with a Monte Carlo approximation and exact gradient computation. Parellizing the evaluation of the acquisition function on a GPU is probably trivial in modern frameworks such as tensorflow etc. The proposed approach is also limited in the sense that most times the evaluations are noisy and it cannot address that setting. It is also not well understood how the constrained setting is addressed when there are feasible observations. In particular, in [21] it is described a modification of expected hyper-volume improvement to account for that. An advanced approximation is also described there to remove the exponential cost of the acquisition function. It is not clear if that approach can be used in the method proposed by the authors. I have read the authors' response. I have decided to keep my score as it is.


Review 2

Summary and Contributions: Multi-objective optimization (MOO) problems involve more than one objective function that are to be minimized or maximized. For non-trivial instances of MOO problems, no unique solution exists that simultaneously optimizes all objectives. In that case, the aim to identify the set of Pareto optimal solutions of the problem. Bayesian Optimization (BO) approaches rely on acquisition functions (AF), to evaluate promising query points for function evaluations. BO approaches for MOO require to define AFs that are applicable to the notion of Pareto optimality. Many existing MOO AFs are limited to one test point per BO iteration due to computational complexity. Therefore, papers propose greedy approaches that evaluate one test point at a time while conditioning on previous outcomes. Usually, these approaches rely on approximations of the acquisition function, i.e. replacing integration over the posterior distribution by posterior means. One well-known AF in the MOO setting is Expected Hypervolume Improvement (EHVI), which extends the Expected Improvement AF from one-dimensional optimization to the formalism of Pareto optimal solutions for MOO. The qEHVI AF is an extension of EHVI to multiple (=q) test points. In its original formulation, evaluating qEHVI requires determining Pareto fronts, which is computationally expensive. In this work, the authors derive a new formula for qEHVI, which replaces this problem by a partitioning problem and the inclusion-exclusion principle for finite measures. The main advantage of this derivation are: - It allows computing exact gradients of Monte Carlo (MC) estimates, which significantly speeds up AF evaluations. - It allows parallelizing computations. - In the greedy setting, it allows to properly integrate over the unobserved points rather than using the posterior mean approximation.

Strengths: One obvious strength of this work is the theoretical foundation of claimed advantages of the approach. The authors provide proofs for two important claims: 1. Under some regularization assumptions of the surrogate model the MC gradients are unbiased estimates to the gradients of the true AF. 2. Instead of applying stochastic approximation methods directly, the authors propose a Sample Average Approximation (SAA) scheme. In the appendix, it is shown that proposed SAA is asymptotically correct. The empirical evaluation is extensive in the setting where the number of partitioning sets, which is required to evaluate the AF is known. The authors compare their method against the following baselines: SMS-EGO, PESMO (Spearmint implementation), analytic EHVI with gradients and a new extension of ParEGO, which allows for parallel evaluation. The authors consider both synthetic and real-world MOO problems. They report the difference of the hypervolume of the approximation of the Pareto front and the true Pareto front at each BO iteration, as well as wall clock time for acquisition function optimization. In a short ablation study, they show the importance of using exact gradients for AF optimization and greedy batch optimization with exact integration for overall optimization performance. Code is provided along with the publication.

Weaknesses: Theoretical Grounding: One drawback of this work is its applicability to noise-free function evaluations only. Theoretical claims of the paper do not hold true in the setting were only noisy function evaluations are available. Implementing this approach in this case would also introduce significant overhead. Relevance: The noise-free assumption additionally limits the applicability of the result to many real-world scenarios where exact evaluations of the optimization targets are not available. Empirical Evaluation: It would be helpful to discuss the role of partitioning algorithm in more detail. For instance, it is stated in the paper that for more than three optimization objectives, the number of partitioning sets, that is required to evaluate the qEHVI AF is unknown. Experiments were run on problems with at most three objectives. Another experiment for the case M>4 could be added to help the reader understand if the partitioning task introduces significant overhead in this case.

Correctness: The claims stated in the main paper seem to be correct. Formally, equation (2) states the inclusion-exclusion principle for finite sets. In general, this principle is not applicable for arbitrary measures, i.e. infinite measures like the Lebesque measure, that is considered in the paper. Nevertheless, since the authors consider only bounded subspaces, the restricted Lebesque measure is indeed finite and the inclusion-exclusion principle is applicable. I would suggest to rephrase equation (2) accordingly.

Clarity: Overall, the paper is well written.

Relation to Prior Work: From my understanding, the idea to compute EHVI (q=1) using partitioning algorithms was already introduced in Yang et. al: Efficient computation of expected hypervolume improvement using box decomposition algorithms. I.e. the current work generalizes this idea to more than one query point per BO iteration and makes use of exact gradient computations. If this is indeed the case, this point should be made clearer in the related work section. From my first read, I got the impression that the idea of applying a partitioning algorithm for AF evaluation was novel.

Reproducibility: Yes

Additional Feedback: Update after rebuttal: I read the other reviews and the authors' response. I still vote for acceptance. --------------------------- The introduction is a bit long. It would be helpful to move some parts from the introduction to a general notation section, in particular the paragraphs about ‘Multi-objective optimization’ and ‘Bayesian Optimization’. Section 3 provides a good introduction to the mathematical concepts used in the paper. Some minor formal issues, completeness of notation in Section 3 should be double-checked: - In Definition 1: appearently y_i are the points contained in P, define [r, y_i] is the hyper-rectangle spanned by r and y_i. - After Definition 2: In the definition of the EHVI acquisition function it is unclear what reference point is used (presumably, the r is equal to the origin if it is omitted in the definition). - In the current wording, Definition 3 does not depend on the reference point r. Presumably, only points that are additionally lower bounded by r are considered. - Equation (4): The input of the qEHVI acquisition function should be multiple evaluation points, not a single point x. Section 4: In the paragraph ‘Optimization via Sample Average Approximation’, are there further modelling assumptions? As far as I understand from the appendix, independency of the GPs is not required. Is that correct?


Review 3

Summary and Contributions: The paper considers the problem of Multi-objective black-box optimization. Multiple useful extensions (constrained, parallel) of the Expected Hypervolume Improvement (EHVI) acquisition function is provided. This is achieved by computing exact gradients of the Monte-Carlo estimator of EHVI acquisition function via auto-differentiation techniques. Experiments show the efficacy of the proposed technique on two synthetic and two real-world benchmarks.

Strengths: Soundness of the claims and theoretical grounding: Most of the claims look correct and are backed by experimental results. Unbiasedness of the gradients of monte-carlo estimator is proved by assuming regularity on the GP's mean and covariance function (although a large part of this analysis is based on Balandat et al. [1]). Significance: In my opinion, the proposed idea will be very useful to the black-box optimization community. An efficient algorithm utilizing latest hardware and auto-differentiation advances is provided in this paper which will increase the utility of EHVI which is a natural acquisition function for multi-objective Bayesian Optimization. References [1]. Maximilian Balandat, Brian Karrer, Daniel R. Jiang, Samuel Daulton, Benjamin Letham, Andrew Gordon Wilson, and Eytan Bakshy. Botorch: Programmable

Weaknesses: There is no particular weakness of the approach. However, I felt that it would be unfair to call the baselines used for comparison as state-of-the-art. There is a considerable amount of work in the last one year on this problem of multi-objective optimization that gives vastly improved performance compared to the considered baselines of ParEGO, SMSego and PESMO (Please see [2], [3], [4], [5]). I would request the authors to compare with few of this approaches (whichever has code available) to further improve the quality of the experimental evaluation. References [2]. Paria, Biswajit, Kirthevasan Kandasamy, and Barnabás Póczos. "A flexible framework for multi-objective bayesian optimization using random scalarizations." arXiv preprint arXiv:1805.12168 (2018). [3]. Suzuki, Shinya, Shion Takeno, Tomoyuki Tamura, Kazuki Shitara, and Masayuki Karasuyama. "Multi-objective Bayesian Optimization using Pareto-frontier Entropy." arXiv preprint arXiv:1906.00127 (2019). [4]. Belakaria, Syrine, Aryan Deshwal, Nitthilan Kannappan Jayakodi, and Janardhan Rao Doppa. "Uncertainty-Aware Search Framework for Multi-Objective Bayesian Optimization." In AAAI, pp. 10044-10052. (2020). [5]. Abdolshah, Majid, Alistair Shilton, Santu Rana, Sunil Gupta, and Svetha Venkatesh. "Multi-objective Bayesian optimisation with preferences over objectives." In Advances in Neural Information Processing Systems, pp. 12235-12245. (2019).

Correctness: Yes, The claims are correct and well-supported by various ablation experiments.

Clarity: The clarity of writing and good exposition is one of the strongest parts of the paper. An excellent review of the EHVI acquisition function is provided in the beginning. Each component of the algorithm is described clearly with experiments to back the claims.

Relation to Prior Work: The paper discusses and contrasts the proposed approach with most of the related work clearly (there are few references missed as given above in [2]-[5], however, this is understandable with the fast-growing pace of research in this area.)

Reproducibility: Yes

Additional Feedback: Update after rebuttal I read the author's rebuttal and keep my review same. It is a really good paper and more work in this direction will be useful for the field.

[Author Response · NeurIPS 2020]

We thank the reviewers for their thorough and insightful comments. We emphasize that our contributions extend EHVI to the important case of parallel constrained optimization. Our approach is practical and highly competitive (including against a novel baseline algorithm, $q$PAREGO, that bests all publicly-available MOBO algorithms). Furthermore, our recent experience with attempting to obtain code from several MOBO authors highlight the value of our work in providing a replicable foundation for advancing cumulative research in this area.

Reviewers raised concerns about limited applicability of $q$EHVI to noisy settings. While EHVI does not account for the noise in the acquisition function (AF) itself, this does not necessarily imply that it would perform poorly on problems with noise. In fact, no previous work on EHVI explicitly accounts for observation noise. Among the many MOBO papers, to our knowledge only PESMO [Hernández-Lobato et al., 2015] and its extensions [Garrido-Merchán and Hernández-Lobato, 2019, 2020], scalarized TS [Paria et al., 2018], and $\epsilon$-PAL [Zuluaga et al., 2016] explicitly handle noise. With the additional page, we plan to address these concerns by including additional evaluation on noisy problems, and we can include a simple extension of $q$EHVI to the noisy setting if the reviewers wish.

In our conclusion, we note that extending $q$EHVI to account for observation noise would be non-trivial, but upon further consideration, there is a straightforward extension inspired by Noisy EI [Letham et al., 2019, Balandat et al., 2019]. The idea behind the approach, which we call $q$NEHVI, is to integrate over the uncertainty in the Pareto frontier (PF) over the previously evaluated points $X_{\text{baseline}}$ by drawing $N$ joint samples from the posterior of $\boldsymbol{f}_t(X_{\text{baseline}}) \sim \mathbb{P}(\boldsymbol{f}(X_{\text{baseline}})|\mathcal{D})$, $t = 1, ..., N$. We prune $X_{\text{baseline}}$ to remove points with zero probability of being Pareto optimal (estimated using MC). For each MC sample, we compute the PF and partition the non-dominated space into disjoint rectangles; this is only required once per BO iteration and can be easily parallelized across multiple processes. Computing the AF is the same as in $q$EHVI, except that we draw $N$ joint samples from $\mathbb{P}(\boldsymbol{f}(X_{\text{baseline}}, \mathcal{X}_{\text{cand}})|\mathcal{D})$ where the $t^{\text{th}}$ sample is conditioned on the original samples $\boldsymbol{f}_t(X_{\text{baseline}})$ to ensure the $t^{\text{th}}$ cached partitioning uses the original samples: $\boldsymbol{f}_t(\mathcal{X}_{\text{cand}}) \sim \mathbb{P}(\boldsymbol{f}(X_{\text{baseline}}, \mathcal{X}_{\text{cand}})|\mathcal{D}, \boldsymbol{f}(X_{\text{baseline}}) = \boldsymbol{f}_t(X_{\text{baseline}}))$. Thus, the only distinction between $q$EHVI and $q$NEHVI is that $q$EHVI uses a partitioning on the PF over observations and $q$NEHVI uses a separate partitioning for the PF over the function values under each MC sample.

We empirically evaluate the performance of all algorithms on a BraninCurrin function where observations have additive, zero-mean, $iid$ Gaussian noise; the unknown standard deviation of the noise is set to be $1\%$ of the range of each objective. Fig 1 shows that $q$EHVI performs favorably in the presence of noise, besting PESMO and TS-TCH, which explicitly account for noise. $q$NEHVI dominates all algorithms, including Noisy $q$PAREGO (described in Appendix E) with respect to log hypervolume difference. Additional problems will be included in the final script, but we have seen similar evidence across a number of informal experiments during testing.

Figure 1: MOBO on the noisy BraninCurrin problem.

$q$NEHVI maintains acceptable optimization wall time. We report the mean and 2 std errors of the wall time per BO iteration in seconds on a CPU: $q$PAREGO 1.8 ($\pm0.2$); $q$NPAREGO 2.1 ($\pm0.2$); EHVI 2.1 ($\pm0.2$); $q$EHVI 2.7 ($\pm0.3$); TS-TCH 13.2 ($\pm0.3$); $q$NEHVI 52.9 ($\pm4.4$); SMS-EGO 87.2 ($\pm5.0$); PESMO 233.12 ($\pm15.02$). We also provide GPU wall times for parallel $q$NEHVI: ($q$=1) 44.9 ($\pm3.4$); ($q$=2) 47.6 ($\pm5.3$); ($q$=4) 82.7 ($\pm8.2$); ($q$=8) 197.5 ($\pm26.4$).

R4 requests additional comparisons. We have included TS-TCH from Paria et al. [2018] in Fig 1 and will include it in all experiments in the final version. We have would liked to include comparisons with other recent work, but Belakaria et al. [2019, 2020], Garrido-Merchán and Hernández-Lobato [2020], and Suzuki et al. [2019], have all graciously declined to share code in response to our request (R4 also mentions Abdolshah et al. [2019], but it is not applicable to the work here).

R1 brought up the handling of constraints. While $q$EHVI cannot use the Pareto dominance rule from Feliot et al. [2016] and maintain differentiability, $q$EHVI gives higher value to candidates that are more likely to be feasible given the candidates have the same MC samples of the objectives, even in the case of no feasible observations.

Regarding R3's comment on scalability with respect to the number of objectives $M$: any exact EHVI computation is exponential in $M$ and therefore is only suitable for moderate $M$ [Yang et al., 2019]. Since $q$EHVI uses the disjoint partitioning for piece-wise integration, it is agnostic to both the partitioning algorithm and the realized partitioning. Therefore, our approach is compatible with more efficient alternative partitioning methods including Yang et al. [2019] and even more scalable approximate partitioning methods such as Couckuyt et al. [2012] (although EHVI computation may no longer be exact with an approximate partitioning).

[Meta-Review · NeurIPS 2020]

*PROS: theoretical guarantees of proposed approach: calculation of unbiased gradients and SAA which is asymptotically correct, extensive evaluation. *CONS: lack of novelty: using an already known acquisition function, expected hyper-volume improvement, which is adapted for quick evaluation in a parallel setting with a Monte Carlo approximation and exact gradient computation Meta-reviewer recommendations: All the reviewers clearly agree on acceptance. I recommend the authors to take into account the reviewers' comments to improve the paper for the final version.